

**Adjusting particle-size distributions to account for**
**aggregation in tephra-deposit model forecasts**
**Larry G. Mastin[1], Alexa R. Van Eaton[1], and Adam J. Durant[2,3]**
[1] [U.S. Geological Survey, Cascades Volcano Observatory, 1300 SE Cardinal Court, Bldg.
10, Suite 100, Vancouver, Washington, USA]
[2] [Section for Meteorology and Oceanography, Department of Geosciences, University of
Oslo, Blindern, 0316 Oslo, Norway]
[3] [Geological and Mining Engineering and Sciences, Michigan Technological University,
1400 Townsend Drive, Houghton, MI 49931, USA]
**Abstract**
Volcanic ash transport and dispersion models (VATDs) are used to forecast tephra deposition
during volcanic eruptions. Model accuracy is limited by the fact that fine ash aggregates,
altering patterns of deposition. In most models this is accounted for by *ad hoc* changes to model
input, representing fine ash as aggregates with density $\rho_{agg}$, and a log-normal size distribution
with median $\mu_{agg}$ and standard deviation $\sigma_{agg}$. Optimal values may vary between eruptions.
To test the variance, we used the Ash3d tephra model to simulate four deposits: 18 May 1980
Mount St. Helens; 16-17 September 1992 Crater Peak (Mount Spurr); 17 June 1996 Ruapehu;
and 23 March 2009 Mount Redoubt. In 158 simulations, we systematically varied $\mu_{agg}$ and
$\sigma_{agg}$, holding $\rho_{agg}$ constant at 600 kg m$^{-3}$. We evaluated the fit using three indices that compare
modeled versus measured (1) mass load at sample locations; (2) mass load versus distance along
the dispersal axis; and (3) isomass area. For all deposits, under these inputs, the best-fit value
of $\mu_{agg}$ ranged narrowly between ~2.1-2.5$\phi$ (0.23-0.18mm), despite large variations in erupted
mass (0.25-50Tg), plume height (8.5-25 km), mass fraction of fine (<0.063mm) ash (3-59%),
atmospheric temperature, and water content between these eruptions. This close agreement
suggests that aggregation may be treated as a discrete process that is insensitive to eruptive style
or magnitude. This result offers the potential for a simple, computationally-efficient



parameterization scheme for use in operational model forecasts. Further research may indicate
whether this narrow range also reflects physical constraints on processes in the evolving cloud.
**1    Introduction**
Airborne tephra is the most wide-reaching of volcanic hazards. It can extend hundreds to
thousands of kilometers from a volcano and impact air quality, transportation, crops, electrical
infrastructure, buildings, water supplies, and sewerage. During eruptions, communities want
to know whether they may receive tephra and how much might fall. Volcano observatories
typically forecast areas at risk by running volcanic ash transport and dispersion models
(VATD). As input, these models require information including eruption start time, plume
height, duration, the wind field, and the size distribution of the falling particles. Of these inputs,
the particle size distribution is perhaps the hardest to constrain.
Particle size (along with shape and density) determines settling velocity, which controls where
particles land in a given wind field. For different eruptions, the total particle-size distribution
(TPSD) can vary. Large eruptions produce more fine ash than small ones for example; and
silicic eruptions produce more than mafic (Rose and Durant, 2009). The TPSD is difficult to
estimate (e.g., Bonadonna and Houghton, 2005); hence estimates exist for only a handful of
deposits. And even in cases where the TPSD is known, that TPSD, entered into a dispersion
model, will not accurately calculate the pattern of deposition (Carey, 1996).
This inaccuracy results from the fact that complex processes, not considered in models, cause
particles to fall out faster than theoretical settling velocities would predict. These processes
include scavenging by hydrometeors (Rose et al., 1995a), gravitational instabilities that cause
dense clouds to collapse *en masse* (Carazzo and Jellinek, 2012; Schultz et al., 2006; Durant,
2015; Manzella et al., 2015), and aggregation, in which ash particles smaller than a few hundred
microns clump into clusters. The rate of aggregation, and the type and size of resulting
aggregates, depend on atmospheric processes such as ice accretion, electrostatic attraction, or
liquid-water binding whose importance varies from place to place.
Although one VATD model, Fall3d, calculates aggregation during transport for research studies
(Folch et al., 2010; Costa et al., 2010), no operational models consider it. Instead, aggregation
is accounted for by either setting a minimum settling velocity in the code (Carey and
Sigurdsson, 1982; Hurst and Turner, 1999; Armienti et al., 1988; Macedonio et al., 1988), or,



in the model input, adjusting particle size distribution by replacing some of the fine ash with
aggregates of a specified density, shape, and size range (Bonadonna et al., 2002; Cornell et al.,
1983; Mastin et al., 2013b). These strategies will probably prevail for at least the next few
years, until microphysical algorithms replace them.
These adjustments are mostly derived from *a posteriori* studies, where model inputs have been
adjusted until results match a particular deposit. It is unclear how well the optimal adjustments
might vary from case to case. For model forecasts during an eruption, we need some
understanding of this variability. This paper addresses this question, using deposits from four
well-documented eruptions. We derive a scheme for adjusting TPSD to account for
aggregation, optimize parameter values to match each deposit, and then see how much these
optimal values vary from one deposit to the next.
**2    Background on the deposits**
The IAVCEI Commission on Tephra Hazard Modeling has posted data from eight well-mapped
eruption deposits, available for use by modeling groups to validate VATD simulations
(http://dbstr.ct.ingv.it/iavcei/). Of these, we focus on eruptions that lasted for hours (not days);
where the TPSD included at least a few percent of ash finer than 0.063mm in diameter; and
where data were available from distal (>35 km) sample locations. Four eruptions met these
criteria: the 18 May 1980 eruption of Mount St. Helens, 16-17 June 1996 eruption of Ruapehu,
and the 16-17 September and 18 August 1992 eruptions of Crater Peak (Mount Spurr), Alaska.
The August Crater Peak eruption was already studied using Ash3d (Schwaiger et al., 2012) and
therefore not included here, reducing the total to three. To these we add event 5 from the 23
March 2009 eruption of Mount Redoubt, Alaska. Although an Ash3d study was made of this
event (Mastin et al., 2013b), aggregation has been unusually well characterized in recent years
(Wallace et al., 2013; Van Eaton et al., in press).
Below are key observations of these events. Deposit maps are shown in Fig. 1, digitized from
published sources.
**1) The 18 May 1980 deposit from Mount St. Helens** remains among the best documented of
any in recent decades (Durant et al., 2009; Sarna-Wojcicki et al., 1981; Waitt and Dzurisin,
1981; Rice, 1981). This 9 hour eruption expelled magma that was dacitic in bulk composition
but contained about 40% crystals and 60% rhyolitic glass (Rutherford et al., 1985). The
eruption start time (1532 UTC) and duration are well documented (Foxworthy and Hill, 1982);
the time-changing plume height was tracked by Doppler radar (Harris et al., 1981) and satellite





(Holasek and Self, 1995) (Table 2). The deposit was mapped within days, before modification
by wind or rainfall, to a distance of ~800 km and to mass load values as low as a few hundredths
of a kilogram per square meter (Sarna-Wojcicki et al., 1981). Estimated volume of the fall
deposit in dense-rock equivalent (DRE) is 0.2 km$^3$ (Sarna-Wojcicki et al., 1981) based on what
fell in the mapped area. A TPSD was estimated by Carey and Sigurdsson (1982) and later by
Durant et al. (2009) to contain about 59% ash <63 um in diameter (Table S1), with a modal
peak in particle size that coincided with the median bubble size of tephra fragments (Genareau
et al., 2012). Some fine ash may have been milled in pyroclastic density currents on the
afternoon of 18 May and in the lateral blast that morning. A secondary maximum in deposit
thickness in Ritzville, Washington (~290 km downwind) was inferred by Carey and Sigurdsson
(1982) to have resulted from fine ash aggregating and falling *en masse*, perhaps as the cloud
descended and warmed to above-freezing temperatures (Durant et al., 2009). Wind directions
that were more southerly at low elevations combined with elutriation off pyroclastic flows in
the afternoon to feed low clouds, producing a deposit that was richer in fine ash along its
northern boundary than in the south (Waitt and Dzurisin, 1981; Eychenne et al., 2015).
Aggregates sampled by Sorem (1982) in eastern Washington consisted mainly of dry clusters
0.250 to 0.500 mm in diameter, containing particles <0.001mm to more than 0.040mm in
diameter, though no aggregates were visible in the fall deposit except at proximal locations (e.g.
Sisson (1995)). The eruption began under clear weather conditions. Clouds increased
throughout the day. Some precipitation in the form of mud rain was noted within tens of
kilometers of the vent (Rosenbaum and Waitt, 1981), probably due to entrainment and
condensation of atmospheric moisture in the rising plume. But no precipitation was recorded
at more distal locations during the event.
**2) The 16-17 September 1991 eruption from Crater Peak, Mount Spurr, Alaska**, was the
third that summer from this vent. The eruption start time (0803 UTC September 17) and
duration (3.6 hours (Eichelberger et al., 1995)) were seismically constrained. The maximum
plume height, measured by U.S. National Weather Service radar (Rose et al., 1995b) increased
for the first 2.3 hours and then fluctuated between about 11 and 14 km above mean sea level
(MSL) until the plume height abruptly decreased at 1110 UTC. The andesitic tephra consisted
of two main types; tan and gray, which were both noteworthy for their low vesicularity (~20-
45%) and high crystallinity (40-100%) (Gardner et al., 1998). The deposit was mapped rapidly
after the eruption (Neal et al., 1995; McGimsey et al., 2001) to a distance of 380 km and mass
loads around 0.050 kg m$^{-2}$. This deposit displays a weak secondary thickness maximum 260-





330 km downwind. Durant and Rose (2009) derived a TPSD for this deposit, estimating about
40% smaller than 0.063 mm. Milling in proximal pyroclastic flows that accompanied this
eruption (Eichelberger et al., 1995) could have contributed fine ash. The eruption occurred at
night under clear skies (Neal et al., 1995).
**3) The 17 June 1996 eruption of Ruapehu** produced a classic weak plume that was modeled
by Bonadonna et a. (2005), Hurst and Turner (1999), Scollo et al. (2008), Liu et al. (2015), and
Klawonn et al. (2014), among others. The main phase involved two pulses, one beginning 16
June at 1910 UTC and lasting 2.5 hours, and the second at 2300 UTC and lasting approximately
1.5 to 2 hours. Ash-laden plumes reached to about 8.5 km altitude above MSL based on satellite
infrared images (Prata and Grant, 2001). The deposit was mapped out to the Bay of Plenty
(190 km), sampled at 118 locations to mass loads less than 0.01 kg m$^{-2}$, and yielded a total mass
of about 0.001 km$^3$ DRE (Bonadonna and Houghton, 2005). Ejecta consisted mainly of scoria
containing 75% glass and 25% crystals, with glass containing about 54 wt% $SiO_2$ (Nakagawa
et al., 1999). A TPSD estimate based on the Voronoi tessellation method (Bonadonna and
Houghton, 2005) suggested that ash <0.063 mm composed only about 3% of the deposit. A
minor secondary thickness maximum was constrained by mapping at about 160 km downwind
(Bonadonna et al., 2005) (Fig. 1c). Although some witnesses at distal locations observed loose,
millimeter-sized clusters falling, no aggregates or accretionary lapilli were present in the
deposit (Klawonn et al., 2014). The eruption was not accompanied by significant pyroclastic
density currents and occurred during clear weather.
**4) Event 5 of the 23 March 2009 eruption of Redoubt Volcano, Alaska** erupted through a
glacier and entrained a variable amount of water into a high-latitude early-spring atmosphere.
It began at 1230 UTC, lasted about 20 minutes on the seismic record (Buurman et al., 2013),
and sent a plume briefly to about 18 km as seen in both National Weather Service NEXRAD
Doppler radar from Anchorage, and a USGS mobile C-band radar system in Kenai, Alaska
(Schneider and Hoblitt, 2013). Within a few days after the eruption, the deposit was mapped
by its contrast with underlying snow in satellite images (NASA MODIS), and sampled for mass
load and particle size distribution at 38 locations, at distances up to ~250 km and mass loads as
low as 0.01 kg m$^{-2}$ (Wallace et al., 2013). During Ash3d modeling of this eruption, Mastin et
al. (2013b) found that wind vectors varied rapidly with both altitude and time, making the
dispersal direction highly sensitive to both the plume height (which varied from ~12 to 18 km
during the 20-minute eruption) and the vertical distribution of mass in the plume. In the deposit,



Wallace et al. (2013) described abundant frozen aggregates with size decreasing with distance
from the vent, from about 10mm at 12 km distance. Schneider et al. (2013) attributed the high
(>50 dBZ) reflectivity of the proximal plume in radar images, and a rapid decrease in maximum
plume height over a period of minutes, to formation and fallout of ashy hail hydrometeors in
the rising column. Van Eaton et al. (2015) combined analysis of the aggregate microstructures
with a 3-D large-eddy simulation to show that the ash aggregates grew directly within the
volcanic plume from a combination of wet growth and freezing, in a process similar to hail
formation.
These eruptions vary from weak (Ruapehu) to strong (Redoubt) plumes, from mid-latitude (St.
Helens, Ruapehu) to high-latitude (Spurr, Redoubt), from dry (Ruapehu) to relatively wet
(Redoubt), from basaltic andesite (Ruapehu) to dacite (St. Helens), and from ~3% to 59% ash
<0.063 mm in diameter. Inferred aggregation processes range from dry (Ruapehu) to wet within
the downwind cloud (St. Helens), to liquid+ice in the rising column (Redoubt).
**3   Methods**
**3.1   The Ash3d model**
We model these eruptions using Ash3d (Schwaiger et al., 2012; Mastin et al., 2013a), an
Eulerian model that calculates tephra transport and deposition through a 3-D, time-changing
wind field. Ash3d calculates transport by setting up a three dimensional grid of cells, adding
tephra into the column of source cells above the volcano, and distributing the mass in the
column following the Suzuki relation (Suzuki, 1983),
$$\frac{dQ_m}{dz} = Q_m \frac{k^2 \left(1 - z/H_v\right) \exp\left(k\left(z/H_v - 1\right)\right)}{H_v \left[1 - \left(1 + k\right) \exp\left(-k\right)\right]}, \tag{1}$$

where $Q_m$ is the mass eruption rate, $H_v$ is plume height above the vent, $z$ is elevation (above the
vent) within the plume, and $k$ is a constant that adjusts the mass distribution.
At each time step, tephra transport is calculated through advection by wind, through turbulent
diffusion, and through particle settling. For wind advection, simulations of Mount St. Helens,
Crater Peak, and Redoubt use a wind field obtained from the National Oceanic and Atmospheric
Administration's (NOAA's) NCEP/NCAR Reanalysis 1 model ("RE1") (Kalnay et al., 1996).
For the Ruapehu simulations we used a local 1-D wind sounding, which gave more accurate
results. The RE1 model provides wind vectors on a global 3-D grid spaced at 2.5° latitude and





longitude, and 17 pressure levels in the atmosphere (1000-10 hPa), updated at 6-hour intervals.
Ash3d calculates turbulent diffusion using a specified diffusivity $D$ (Schwaiger et al., 2012, eq.
4). $D$ is set to zero for simplicity, though later we show the effect of different values of $D$.
Settling rates are calculated using relations of Wilson and Huang (1979) for ellipsoidal particles.
Wilson and Huang define a particle shape factor $F \equiv (b + c)/2a$, where $a$, $b$, and $c$ are the semi-
major, intermediate, and semi-minor axes of the  ellipsoid respectively.  Wilson and Huang
measured $a$, $b$, and $c$ for 155 natural pyroclasts.  The average $F$ of their measurements was 0.44,
which we use in our model.  For aggregates we use $F$=1.0 (round aggregates).
Other model inputs include the extent and nodal spacing of the model domain; vent location
and elevation; the eruption start time, duration, plume height, erupted volume, diffusion
coefficient $D$, and a series of particle size classes and associated densities.  The size classes
may represent either individual particles or aggregates.  These input values are given in Tables
1 and 2.

## 3.2    Adjusting particle size distributions to account for aggregation

The TPSD used to model these four eruptions are listed in Table S1 and illustrated in Fig. 2.
We aim to adjust the TPSD in our model to better match the mapped deposits.  In doing so, we
assume that some fraction ($m_{agg}$) of ash smaller than some size $\phi_p^{max}$ collects into clusters having
a density $\rho_{agg}$ and Gaussian size distribution of mean $\mu_{agg}$, and standard deviation $\sigma_{agg}$.  For
deposit modeling, we ignore the small fraction of the erupted mass that goes into the distal
cloud, typically a few percent (Dacre et al., 2011; Devenish et al., 2012).   In the Appendix we
briefly review aggregation processes. We offer the following parameterization scheme:
For $\phi$ >=4, all ash aggregates
For $\phi$ <=2, no ash aggregates.
For 4>$\phi$>2, the mass fraction that aggregates varies linearly with $\phi$ from 1 (when $\phi$=4) to 0
(when $\phi$=2).
Based on this scheme, particle sizes that aggregate are depicted as gray bars in Fig. 2.





### 3.3    Statistical measures of fit

For each eruption, we have done a series of model simulations, first using the TPSD without considering aggregation, and then systematically varying $\sigma_{agg}$ and $\mu_{agg}$ to include the effects of aggregation. We compare the resulting deposit with the mapped deposit using three methods presented in Table 3. Each has advantages and disadvantages.

1) **The point-by-point index** $\Delta^2$ compares model results with sample data collected at specific locations (dots, Fig. 1). It offers the advantage that the comparison is made directly with measured values, not with interpreted or extrapolated contours of data. But $\Delta^2$ values are dominated by differences in proximal locations where mass per unit area is greatest; and values of $\Delta^2$ can be influenced by errors in the wind field, which cannot be adjusted in the model.

2) **The downwind thinning index** $\Delta^2_{downwind}$, compares modeled mass per unit area along the downwind dispersal axis with values expected at that distance based on a trend line drawn from field measurements (Fig. 3). The comparison is not made directly with measured values (a disadvantage). However the method does not suffer the limitation of over-weighting proximal data. And, more importantly, it still provides a useful comparison when wind errors cause the modeled dispersal axis to diverge from the mapped one.

3) **The isomass area index** $\Delta^2_{area}$ compares the area within modeled and mapped isomass lines. It is based on traditional plots of the log of isopach thickness versus square root of area (Pyle, 1989; Fierstein and Nathenson, 1992; Bonadonna and Costa, 2012), which are assumed to accurately depict the areal distribution of tephra while minimizing the effects of 3-D wind on the distribution (Pyle, 1989). Fig. 4 shows plots for our four eruptions, using the log of isomass rather than isopach thickness to avoid problems introduced by varying deposit density.

The index $\Delta^2_{area}$ is assumed to be insensitive to effects of wind (an advantage). However, model results are compared with isopach lines that are interpretive and may not be well constrained, depending on the distribution and number density of sample locations.

### 3.4    Sensitivity to various input values

We ignore complex, proximal fallout and concentrate on medial to distal areas, about 100 to ~500 km downwind for example at Mount St. Helens. There, under the average wind speed (15.1 m s$^{-1}$) that existed below about 15 km, tephra falling from 15km at average settling



velocities of 0.4-1.5 m s$^{-1}$ would deposit within this range (Fig. 5a). Tephra falling at 0.66-0.78
m s$^{-1}$ would land 290-340 km downwind, the distance of the secondary maximum at Ritzville.
A wide range of aggregate diameters $d$ could fall at this rate depending on density $\rho_{agg}$ (Fig.
5b). For simplicity, we hold $\rho_{agg}$ constant at 600 kg m$^{-3}$, toward the middle of the observed
range of aggregate densities (~50-1600 kg m$^{-3}$ (Sparks et al., 1997, Table 16.1; Taddeucci et
al., 2011)).
Other factors listed below can also affect the results.
***Aggregate shape.*** Aggregate shape can strongly affect the settling velocity and thus where
deposits fall, as illustrated in Fig. 6. For simplicity, we use round aggregates ($F$=1.0).
***Suzuki k***. Simulations of Mount St. Helens (Fig. 7) show that increasing the Suzuki factor from
4 to 8 increases the prominence of a secondary thickness maximum. But at $k$>~8, the proximal
deposit becomes unrealistically thin. Our simulations use $k$=8 to replicate the known prominent
secondary thickening while minimizing unrealistic thinning of proximal deposits.
***Aggregate size***. The transport distance is highly sensitive to aggregate size. Reducing
aggregate diameter $d$ from 0.250 to 0.217 to 0.189 mm increases transport distance at Mount
St. Helens from 300 to 366 to 448 km respectively (Fig. 5a). In simulations that use a single,
dominant aggregate size, these variations produce conspicuous changes in the location of a
secondary maximum (Fig. 8). Decreasing size also decreases the percent of erupted mass lands
in the mapped area: from 70% to 53% to 39% for $d$=0.165, 0.143, and 0.125mm respectively.
Our simulations limit $\mu_{agg}$ to values of 1.8-3.1$\phi$ (0.287-0.117mm), and $\sigma_{agg}$ to 0.1-0.3$\phi$, to
ensure that most deposits fall in the region of interest.

**4    Results**
We ran simulations at $\mu_{agg}$ =1.8, 1.9, 2.0 . . . 3.1$\phi$, and $\sigma_{agg}$ 0.1, 0.2, and 0.3$\phi$. The latter used
1, 3, and 5 aggregate size classes respectively, in each simulation, with the percentage of fine
ash assigned to each bin given in Table 4. Our calculations of $\Delta^2$ and $\Delta^2_{downwind}$ only included
sample points whose downwind distance lay within the range indicated by the trend lines in
Fig. 3.



Figure 9 shows contours of $\Delta^2$, $\Delta^2_{downwind}$, and $\Delta^2_{area}$ as a function of $\sigma_{agg}$ and $\mu_{agg}$ for each of
these four deposits. Values are given in Tables S3-S6. Although the three indices compare
different features of the deposit, they provide roughly similar optimal values of $\sigma_{agg}$ and $\mu_{agg}$.
For Mount St. Helens, for example, the best-fit value of $\mu_{agg}$ is about 2.3ϕ using $\Delta^2$ (Fig. 9a),
2.5ϕ using $\Delta^2_{downwind}$ (Fig. 9b), and 2.6ϕ using $\Delta^2_{area}$ (Fig. 9c). The fit does not depend very
strongly on $\sigma_{agg}$ but appears slightly better at higher values. For Crater Peak, optimal $\mu_{agg}$
values are 2.3ϕ, 2.2ϕ, and 1.9ϕ respectively. For Ruapehu they are about 2.1-2.4ϕ (poorly
constrained), 2.2ϕ, and 2.3ϕ. For both Crater Peak and Ruapehu, the fit is also insensitive to
$\sigma_{agg}$, though slightly better at higher values for Ruapehu using $\Delta^2_{area}$ (Fig. 9i). For Redoubt,
optimal values are disparate: $\mu_{agg}$ =2.1-2.2ϕ, 2.3ϕ, and <1ϕ respectively. The Redoubt deposit
is least constrained by field data and the most difficult to match due to the complex wind
conditions.
Figures 10-13 show results for each of these eruptions using $\mu_{agg}$ =2.4ϕ (0.29mm) and $\sigma_{agg}$
=0.3ϕ. The sizes of particles and aggregates used to generate these figures is given in Table
S2. For all deposits these values are close to optimal, depending on which criterion is used.
Similar figures for other values of $\mu_{agg}$ and $\sigma_{agg}$ are provided as Figs. S005-S172.
Figures S001-S004 show simulations using the original particle-size distribution, with no
aggregation. Tephra fall beyond a few tens of kilometers is strongly underestimated in all these
runs, especially for the three eruptions that contain more than a few percent fine ash. Values
of $\Delta^2$, $\Delta^2_{downwind}$, and $\Delta^2_{area}$ are also higher than most simulations that use aggregates (Table S3-
S6). For Mount St. Helens, Crater Peak, Ruapehu, and Redoubt, the percentages of the erupted
mass landing in the mapped area are very low: 29%, 42%, 88%, and 59% respectively.
Optimal aggregates obtained from our study are similar in size but denser than those found
optimal by Cornell et al. (1983) for the Campanian Y-5 ($\mu_{agg}$ =2.3ϕ, $\rho_{agg}$=200 kg m$^{-3}$). The
unknown wind field during the prehistoric Campanian Y-5 eruption makes it difficult to
compare Cornell et al.'s optimal value to the results here. Folch et al. (2010) matched the
Mount St. Helens deposit using a similar aggregation scheme, but with aggregates of density



400 kg m$^{-3}$ (compared with our 600 kg m$^{-3}$) and diameter of 0.2-0.3mm (compared with our
~0.2mm). Their results are broadly consistent with ours.

### 4.1   Mount St. Helens

For the Mount St. Helens case, the modeled deposit follows a dispersal axis (solid black line,
Fig. 10a) that matches almost exactly with the mapped one (dashed line). The agreement
reflects both the faithfulness of the numerical wind field to the true one and the appropriateness
of other inputs, such $k$, that influence dispersal direction. The measured mass loads in Fig. 10a,
indicated by the color of markers, agree reasonably well with modeled mass loads indicated by
colors of the contour lines, except along the most distal transect, where modeled loads are
essentially zero while measured loads are about 10$^{-1}$ kg m$^{-2}$. Figure 10b shows that modeled
and measured mass loads generally agree within a factor of three or so, except for those same
distal, low-mass-load measurements, to the lower left of the legend label (those where modeled
values are truly zero do not show up on this plot). Figure 10c shows that the modeled mass
load (black line with dots) contains a secondary thickening at about the same location mapped
(dashed line). However, the modeled mass load is consistently less than measured, especially
at the most distal sites. In Figure 10d, the log of modeled mass load versus square root of area
shows reasonable agreement with mapped values until mass loads are less than about 1 kg m$^{-2}$,
where they diverge.
Notably, modeled mass loads somewhat underestimate the measured values along the dispersal
axis in Fig. 10c. The underestimate reflects the fact that the input erupted volume of 0.2 km$^3$
DRE (Table 1) was based on estimates by Sarna-Wojcicki et al. (1981) of what lay within the
mapped area in Fig. 10a; yet only about 79% of the modeled mass landed within this area.
Reducing the mean aggregate size to 2.7$\phi$ (0.153mm, Figs. S032-S034) improves the fit
somewhat along distal transect near the dispersal axis but not along the entire transect length.
And the finer size moves the secondary maximum too far east and reduces the percentage
deposited to 50-60%.
In Fig. 10a, the modeled deposit is also narrower than the mapped one. Adding turbulent
diffusion, with a diffusivity $D$ of about 3×10$^2$ m$^2$ s$^{-1}$ (Fig. 14) visually improves the fit, and was
likely important during this eruption due to high crosswind speeds that increased entrainment
(Degruyter and Bonadonna, 2012; Mastin, 2014). Ignoring turbulent diffusion decreases run





time by ~3x, from ~30 to 10 minutes for operational runs, and is a reasonable compromise
under operational conditions.

## 4.2   Crater Peak (Mount Spurr)

At Crater Peak (Mount Spurr), results in Fig. 11a also show good agreement between the
modeled dispersal axis and the mapped one (which is constrained by fewer sample locations
than the Mount St. Helens case). The isomass lines in this plot are jagged and irregular due to
effects of topography in this mountainous region. The modeled location of secondary
thickening in Fig. 11c agrees with the mapped location, about 250-300km downwind. Although
Fig. 11c shows a tendency to underestimate the mass load along the dispersal axis, there is less
tendency to underestimate mass load in the most distal locations as occurred at Mount St.
Helens. In Fig. 11d, the areas covered by modeled isomass lines are comparable to the mapped
values, down to mass loads approaching 0.1 kg m$^{-2}$.

## 4.3   Ruapehu

For Ruapehu (Fig. 12), simulations using the NCEP Reanalysis 1 numerical winds produced an
odd double dispersal axis whose average did not correspond well with the mapped direction of
dispersal (Fig. 1c). To improve the fit we used the 1-D wind sounding provided for this eruption
at the IAVCEI Tephra Hazard Modeling Commission web page (http://dbstr.ct.ingv.it/iavcei/).
Use of a 1-D wind sounding seems justified in this case because this deposit covers a smaller
area than the others, making a 3-D wind field less important in calculating transport. The
resulting dispersal axis (Fig. 12a) agrees with the mapped one out to about 140 km distance,
beyond which it strays eastward, reaching the coast, 180 km downwind, about 10 km east of
the mapped axis. This slight difference is enough to cause misfits in point-to-point comparisons
at measured mass loads of ~10$^{-1}$ kg m$^{-2}$ (Fig. 12b).
The modeled mass load along the dispersal axis (Fig. 12c) agrees with measurements to about
60-90 km distance. At 100-200 km, modeled values level off and show a hint of secondary
thickening at ~180 km, in agreement with the mapped deposit (Fig. 1c and 11c), although the
mapped secondary thickening is more prominent.
A large discrepancy is also apparent at distances of less than 60 km, where mass load along the
dispersal axis (Fig. 12c) and the area covered by thick isomass lines (Fig. 12d) is greater than
the mapped deposit. The implication is that too much mass is dropping out proximally in the




model.  Underestimates of isomass area at $<=10^{-1}$ kg m$^{-2}$ (Fig. 12d) also show that too little is
falling distally.  Simulations (not shown) that raise the plume height or increase $k$ to concentrate
more mass high in the plume do not improve the fit.  The discrepancy may reflect the coarse
TPSD—50% of which is coarser than 1mm (compared with 2%, 12%, and 8% for the other
three deposits in Table S1).  An additional simulation used the TPSD derived from technique
B of Bonadonna and Houghton (2005) (Table S1), which divides the deposit into arbitrary
sectors, and calculates a weighted sum of the size distributions in each sector following Carey
and Sigurdsson (1982).  Technique B yields a finer average particle size than technique C,
which uses Voronoi tessellation to sectorize the deposit. But the finer particle size of the
technique B TPSD does not improve the fit (Fig. S173).  Further exploration of this discrepancy
is beyond the scope of this paper; but other possible causes could include release of different
particle sizes at different elevations, or complex transport in the bending of the weak plume that
can't be accommodated in this model.
A second, smaller discrepancy is that the modeled deposit is narrower than the mapped one
(Fig. 1c).  As at Mount St. Helens, deposit widening due to cross flow entrainment is likely.
Increases in entrainment resulting from crossflow is widely known to both increase plume width
and decrease its height for a given eruption rate (Briggs, 1984; Hoult and Weil, 1972; Hewett
et al., 1971; Woodhouse et al., 2013).  Adding turbulent diffusion, we get a visually improved
fit when $D=\sim 3\times 10^3$ m$^2$ s$^{-1}$ (Fig. 15), consistent with findings by Bonadonna et al. (2005) based
on the rate of downwind widening of isomass lines.  This diffusivity is also similar to the visual
best-fit value for Mount St. Helens (Fig. 14).
Despite the uncertainty in TPSD, simulations that systematically vary $\mu_{agg}$ and $\sigma_{agg}$ fit best in
Figs. 9g, h, and i when $\mu_{agg}$ is about 2.2 to 2.4.  Results similar to those presented in Fig. 12c
use other values of $\mu_{agg}$ (Figs. S089-S130) and show a secondary maximum migrating
downwind as $\mu_{agg}$ increases, coming into agreement with the mapped distance at $\mu_{agg}=2.2$ to
2.4$\phi$ (0.19-0.22mm), where errors in Fig. 9g, h, and i are lowest.

## 4.4   Redoubt

This deposit is the second smallest in our group, the least well-constrained by sampling, and
the only one in our group not known to include a secondary thickness maximum.  Mastin et al.
(2013b) modeled this deposit using numerical winds from the North American Regional



Reanalysis model (Mesinger et al., 2006). During that eruption, the winds at 0-4 km, 6-10, and
>10 km elevation were directed toward the northwest; north, and northeast respectively, with
the highest speeds at 6-10 km. Mastin et al. found that the modeled cloud developed a north-
oriented, northward migrating wishbone shape with the west prong at low elevation and the east
prong at high elevation. Mastin et al. also found that the modeled dispersal axis and the mass
load distribution roughly agreed with mapped values for a plume height of 15km, $k$=8, and a
particle size adjustment that involved taking 95% of the fine ash (<0.063mm) and distributing
it evenly among the coarser bins. In this study we use the same plume height and $k$ value, a
different wind field (RE1), and explore a different parameterization for particle aggregation.
In Fig. 13a, the modeled dispersal axis diverges about 20° westward from the mapped axis. We
do not correct this divergence by adjusting mass height distribution, since the optimal values of
$\mu_{agg}$ and $\sigma_{agg}$ can still be obtained from $\Delta^2_{downwind}$, and $\Delta^2_{area}$. As with the Crater peak (Spurr)
simulations, the isomass lines are jagged and patchy; an artifact of high relief. (The most distal
sample location lies at 4.3 km elevation on the west shoulder of Mount McKinley). Although
the value of $\mu_{agg}$ (2.4ϕ, 0.20mm) portrayed in Fig. 13 is close to optimal in Fig. 9j, many sample
points do not plot in Fig. 13b because modeled mass load is zero. And most values of $\Delta^2$ are
high—0.99, largely because of the disparity in axis dispersal directions and the consequent fact
that sample points lie outside the modeled deposit. The reason that $\Delta^2$ shows a clear minimum,
around $\mu_{agg}$=2.4ϕ (0.20mm) in Fig. 9j, is apparent from Figs. S131-S172 which show that, as
$\mu_{agg}$ decreases in size, the modeled deposit extends farther north and takes a clear turn to the
northeast, overlapping more with the mapped deposit. These figures also illuminate why
$\Delta^2_{downwind}$ is optimal at $\mu_{agg}$=2.3; because modeled and mapped loads come into best agreement
along the dispersal axis for aggregates of this size. $\Delta^2_{area}$ is optimized at $\mu_{agg}$<1 because the
area of the 1 kg m$^{-2}$ isomass diverges below the mapped value, and the area of the 0.01 kg m$^{-2}$
isomass diverges above observed, as aggregate size increases. The isomass lines are drawn
based on sparse data and are the least reliable of the datasets used in this comparison.
**5   Discussion and Conclusions**
The overall derived values of $\mu_{agg}$ have a narrow range between ~2.1-2.5ϕ (0.18-0.23mm),
despite large variations in erupted mass (0.25-50×Tg), plume height (8.5-25 km), mass fraction




of fine (<0.063mm) ash (3-59%), atmospheric temperature, and water content between these
eruptions. The value of this narrow range depends strongly on other inputs, such as particle
density, shape factor, and Suzuki factor. But, holding those factors constant, the similarity in
this range between these four eruptions is noteworthy.
The overall agreement in modeled mean aggregate size ($\mu_{agg}$) suggests that accelerated fine-
ash deposition may be treated as a discrete process, insensitive to eruptive style or magnitude.
It seems unlikely that these varied eruptions would produce aggregates of the same size, density,
and morphology. A combination of processes removed ash. Our approach captures these
processes implicitly, ignoring the microphysics.
What sort of processes could evolve in the cloud? Some possibilities are illustrated in Fig. A1.
The evolution starts with ejection of particles from that vent whose size ranges from microns
to meters. For an eruption having the TPSD of Mount St. Helens, the rising plume would have
contained $10^6$-$10^8$ particles per cubic meter with diameter between 10-30 μm that collided with
larger particles hundreds of thousands of times per second. High collision rates and the
availability of liquid water in the plume would have led to rapid aggregation. Freezing of liquid
water and riming would have shifted the maximum possible size of aggregates towards mm to
cm sizes. Mud rain, observed falling at Mount St. Helens (Waitt, 1981) and ice aggregates
collected near the vent at Redoubt (Van Eaton et al., in press), are evidence of these processes.
In the downwind cloud particle concentrations were lower, turbulence was less intense, a
smaller range of particle sizes existed, and, for all four eruptions, atmospheric temperatures
near the plume top were well below freezing (Table 5), leading to presumably slow aggregation
rates. However, at least two other processes may help settle ash from downwind clouds. One
is gravitational overturn. Experiments (Carazzo and Jellinek, 2012) have observed that fine ash
settles toward the bottom of ash clouds as they expand and move downwind, accumulating
gravitationally unstable particle boundary layers that eventually overturn and cause the entire
air mass to settle rapidly. At Eyjafjallajökull in 2010, gravitational convective instabilities
formed within 10km of the vent, presumably as a result of accumulation of coarse ash over a
period of minutes (Manzella et al., 2015). The development of fine-ash particle boundary layers
presumably takes longer, perhaps hours, although the underlying processes remain a subject of
active research.





A second process is hydrometeor growth. In some cases, magmatic and (or) externally-derived
water in the eruption cloud may condense on ash particles and initiate hydrometeor growth.
Both hydrometeor growth and gravitational overturn have been suggested to produce the
mammatus clouds that developed in mid-day over central Washington on 18 May 1980 and
signaled mass settling (Durant, 2015; Durant et al., 2009; Carazzo and Jellinek, 2012).
Mammatus descent rates are typically meters per second (Schultz et al., 2006), much faster than
the settling rate of individual ash particles ($<0.1$ m s$^{-1}$) or even of ash aggregates ($<\sim1$ m s$^{-1}$,
Fig. 5).
The extent to which these processes operated at Crater Peak, Ruapehu, and Redoubt is
unknown. Cloud structures were not observed during the nighttime eruptions of Redoubt and
Crater Peak (Spurr). And although virga-like structures can be seen in some near-vent photos
of Ruapehu (Bonadonna et al., 2005, Fig. 9a), we have seen no documentation of such
instabilities farther downwind.
For operational forecasting, these mechanisms cannot be considered in any case, because no
operational model has the capability to resolve these processes. The fact that these eruptions
can all be reasonably modeled using similar inputs for aggregate size is convenient, even if the
model does not calculate the processes involved. The agreement suggests that model forecasts
can still be useful during the coming years. Future work will focus on the development of more
sophisticated algorithms that account for cloud microphysics.
**6   Appendix**
The rate and extent of ash aggregation are sensitive to changes in both eruptive
conditions and background meteorology. Despite the complexity of the process, field studies
and laboratory experiments have highlighted key spatial and temporal controls. For example,
large aggregates, including frozen accretionary lapilli, tend to form near the volcanic source
and are particularly abundant in phreatomagmatic eruption deposits (Van Eaton et al., 2015;
Brown et al., 2012; Houghton et al., 2015). These are associated with precipitation-forming
processes occurring as particles collide in moist, turbulent updrafts rising above the volcanic
vent or ground-hugging density currents (Fig. A1). Field measurements indicate that near-
source aggregates commonly exceed 1 cm diameter (Wallace et al., 2013; Swanson et al., 2014;
Van Eaton and Wilson, 2013). In contrast, the low-density aggregates that produced the
Ritzville Bulge, 230 km downwind from Mount St. Helens, are thought to have been triggered
by mammatus cloud instabilities (Durant et al., (2009). As the cloud descended to warmer



atmospheric levels, the increasing proportion of liquid water increased the rate of aggregation
and fallout (red line, Fig. A1). These types of distal aggregates tend to be smaller than a
millimeter, forming in the downwind cloud up to hundreds of kilometers from source (Sorem,
1982; Dartayat, 1932).
Liquid water also influences aggregate morphology, density, and rate of formation. Laboratory
experiments have shown that wet ash (>10-15 wt.% liquid water) rapidly produces dense, sub-
spherical pellets, whereas drier conditions lead to low-density, electrostatically-bound clusters
(Schumacher and Schmincke, 1995; James et al., 2002; Van Eaton et al., 2012). Furthermore,
aggregation is a highly size-selective process – smaller particles (<0.25mm) have a much
greater likelihood of sticking (Gilbert and Lane, 1994; Schumacher and Schmincke, 1995; Van
Eaton et al., 2012). In this study, we do not attempt to address the detailed mechanisms of
aggregation, but consider the bulk impact on downwind deposits for practical applications in
ash dispersal forecasting.
**Author contributions**
L. Mastin conceived the study, did the model simulations and wrote most of the paper. A. Van
Eaton provided advice on aggregation processes and wrote the appendix. A. Durant provided
the data for Mount St. Helens and Crater Peak, and advice on aggregation processes that
occurred during those two eruptions.
**Acknowledgments**
We are grateful to the IAVCEI Commission on Tephra Hazard Modeling for posting data on
key eruptions that could be used for this study.



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



**Tables**
Table 1: Input parameters for simulations. Vent elevation is given in kilometers above mean
sea level.

| PARAMETER(S) | MOUNT ST. HELENS | SPURR | RUAPEHU | REDOUBT |
|---|---|---|---|---|
| MODEL DOMAIN | 42-49°N 124-110°W 0-35 km asl | 59-64°N 155.6-141.4°W 0-17 km asl | 39.5-37.5°S 175-177°E 0-12km asl | 60-64°N 155-145°W 0-20km asl |
| VENT LOCATION | 122.18°W 46.2°N | 152.25°W 61.23°N | 175.56°E 39.28°S | 152.75°W 60.48°N |
| VENT ELEVATION (KM) | 2.00 | 2.30 | 2.80 | 2.30 |
| NODAL SPACING | 0.1° horizontal 1.0 km vertical | 0.1° horizontal 1.0 km vertical | 0.025° horizontal 0.5 km vertical | 0.07° horizontal 1.0 km vertical |
| ERUPTION START DATE (UTC) (YYYY.MM.DD) | 1980.05.18 | 1992.09.17 | 1996.06.16 1996.06.17 | 2009.03.23 |
| START TIME (UTC) | 1530 | 0803 UTC | 2030 UTC 0200 UTC | 1230 UTC |
| PLUME HEIGHT, KM ASL | See Table 2 | 13 | 8.5 | 15 |
| DURATION, HRS | See Table 2 | 3.6 | 4.5 2.0 | 0.33 |
| ERUPTED VOLUME KM$^3$ DRE | 0.2 (total) | 0.014 | 0.000643 0.000357 | 0.0017 |
| DIFFUSION COEFFICIENT $D$ | 0 | 0 | 0 | 0 |
| SUZUKI CONSTANT $K$ | 8 | 8 | 8 | 8 |
| PARTICLE SHAPE FACTOR $F$ | 0.44 | 0.44 | 0.44 | 0.44 |
| AGGREGATE SHAPE FACTOR $F$ | 1.0 | 1.0 | 1.0 | 1.0 |







Table 2: Time series of plume height and total erupted volume used in model simulations of
the Mount St. Helens ash cloud. *H*=plume height in km above sea level (a.s.l.), V=erupted
volume in million cubic meters dense-rock equivalent (DRE). The time series of plume height
approximates that measured by radar (Harris et al., 1981). We calculated a preliminary eruptive
volume for each eruptive pulse using the duration and the empirical relationship between plume
height and eruption rate (Mastin et al., 2009). This method underestimated the eruptive volume,
as noted in previous studies (Carey et al., 1990). Hence we adjusted the volume of each pulse
proportionately so that their total equals the 0.2 $km^3$ DRE estimated by Sarna-Wojcicki et al.
(1981). For the last two eruptive pulses, start times in UTC, marked with asterisks, are on 19
May in UTC time. All other start times are on 18 May.

**Plume height (*H*), duration (*D*) and volume (*V*)**

| start | | *D* | *H* | *V* |
|-------|-------|-----|-------|-------------------|
| PDT | UTC | *min* | *km asl* | $\times 10^6$ m$^3$ DRE |
| 8:30 | 1530 | 30 | 25 | 3.247 |
| 9:00 | 1600 | 36 | 15.3 | 0.077 |
| 9:36 | 1636 | 54 | 13.7 | 0.356 |
| 10:30 | 1730 | 45 | 15.3 | 0.502 |
| 11:15 | 1815 | 30 | 16.1 | 0.426 |
| 11:45 | 1845 | 42 | 17.4 | 0.615 |
| 12:27 | 1927 | 48 | 17.4 | 0.615 |
| 13:15 | 2015 | 60 | 14.6 | 0.183 |
| 14:15 | 2115 | 45 | 14.7 | 0.535 |
| 15:30 | 2230 | 60 | 15.8 | 0.691 |
| 16:30 | 2330 | 60 | 19.2 | 0.700 |
| 17:30 | 0030* | 60 | 7.7 | 1.945 |
| 18:30 | 0130* | 60 | 6.2 | 0.020 |




Table 3. Statistical measures of fit used in this paper

| Name | Formula | Explanation |
|---|---|---|
| **Point-by-point method** | $\Delta^2 = \left[ \dfrac{\sum\limits_{i=1}^{N}\left(m_{m,i}-m_{o,i}\right)^2}{\sum\limits_{i=1}^{N}m_{o,i}^2} \right]^{1/2}$ | The mass load $m_{o,i}$ observed at each sample location $i$ is compared with modeled mass load $m_{m,i}$ at the same location. Squared differences are summed to the total number of sample points $N$, and normalized to the sum of squares of the observed mass loads. |
| **Downwind thinning method** | $\Delta^2_{downwind} = \dfrac{1}{M}\sum\limits_{j=1}^{M}\left(\log\left(m_{m,j}/m_{o,j}\right)\right)^2$ | The log of modeled mass load $m_{m,j}$ at a point $j$ on the dispersal axis, is compared with the observation-based value $m_{o,j}$ expected at that location based on a trend line drawn between field measurements along the axis (Fig. 7). Differences between $m_{m,j}$ and $m_{o,j}$ are calculated on a log scale, squared, and summed. |
| **Isomass area method** | $\Delta^2_{area} = \left[ \dfrac{\sum\limits_{i=1}^{L}\left(A_{m,i}-A_{o,i}\right)^2}{\sum\limits_{i=1}^{L}A_{o,i}^2} \right]^{1/2}$ | This method calculates the area $A_{m,i}$ of the modeled deposit that exceeds a given mass load $i$ by summing the area of all model nodes that meet this criterion. It then takes the difference between $A_{m,i}$ and the area $A_{o,i}$ within same isomass line mapped from field observations. The sum of the squares of these differences, normalized to the sum of the squared mapped isopach areas, gives the index $\Delta^2_{area}$. |




Table 4: percentage of fine ash assigned to different size bins for different values of $\sigma_{agg}$.

| | $\mu_{agg}$ -0.2 | $\mu_{agg}$ -0.1 | $\mu_{agg}$ | $\mu_{agg}$ +0.1 | $\mu_{agg}$ +0.2 |
|---|---|---|---|---|---|
| $\sigma_{agg}$ =0.1 | | | 100% | | |
| $\sigma_{agg}$ =0.2 | | 25% | 50% | 25% | |
| $\sigma_{agg}$ =0.3 | 10% | 20% | 40% | 20% | 10% |







Table 5:  Atmospheric temperature profiles during the eruptions at Mount St. Helens, Crater
Peak (Spurr), Ruapehu, and Redoubt volcanoes.  Profile for Mount St. Helens is for 18 May
1980, 1800 UTC, interpolated to the location of Ritzville, Washington (47.12°N, 118.38°W).
For Crater Peak (Spurr) the profile is for 17 September 1992, 1200 UTC, interpolated to the
location of Palmer, Alaska (61.6°N, 149.11°W).  For Ruapehu the temperature profile is for
17 June 1996, 0000 UTC, interpolated to the location of Ruapehu.  For Redoubt the sounding
was for 23 March 2009, 1200 UTC, at 62°N, 153°W.  All soundings were taken from using
RE1 reanalysis data at http://ready.arl.noaa.gov/READYamet.php.    For Mount St. Helens,
the freezing elevation was also checked using data from the North American Regional
Reanalysis (NARR) model (Mesinger et al., 2006), available at the same NOAA site, and
found to be 3.3 km, similar to that given below by the RE1 model.

|  | Mount St. Helens | | Crater Peak (Spurr) | | Ruapehu | | Redoubt | |
| --- | --- | --- | --- | --- | --- | --- | --- | --- |
| p (hPa) | z (m) | T (C) | z (m) | T (C) | z (m) | T (C) | z (m) | T (C) |
| 10 | 31,381 | -39.9 | 31,137 | -41.8 | 30,632 | -54.9 | 30,179 | -61.9 |
| 20 | 26,713 | -47.5 | 26,535 | -51.0 | 26,239 | -57.9 | 25,891 | -62.1 |
| 30 | 24,067 | -52.1 | 23,920 | -54.4 | 23,673 | -56.6 | 23,385 | -61.3 |
| 50 | 20,786 | -55.7 | 20,660 | -55.5 | 20,441 | -57.1 | 20,185 | -57.6 |
| 70 | 18,646 | -55.8 | 18,515 | -55.6 | 18,307 | -56.4 | 18,049 | -55.1 |
| 100 | 16,377 | -55.4 | 16,241 | -55.3 | 16,041 | -56 | 15,759 | -53.1 |
| 150 | 13,782 | -55.1 | 13,646 | -56.0 | 13,439 | -54.2 | 13,133 | -51 |
| 200 | 11,962 | -58.3 | 11,833 | -58.9 | 11,613 | -58.6 | 11,255 | -50.4 |
| 250 | 10,552 | -53.4 | 10,412 | -51.3 | 10,214 | -58.3 | 9,814 | -54.7 |
| 300 | 9,355 | -44 | 9,200 | -41.0 | 9,057 | -53.4 | 8,652 | -55.5 |
| 400 | 7,355 | -28.5 | 7,174 | -25.0 | 7,151 | -38.9 | 6,764 | -41.9 |
| 500 | 5,716 | -16.4 | 5,519 | -15.5 | 5,576 | -26.7 | 5,225 | -33.9 |
| 600 | 4,318 | -6.9 | 4,126 | -10.2 | 4,231 | -15.5 | 3,929 | -27.4 |
| 700 | 3,100 | 0.1 | 2,929 | -6.7 | 3,049 | -8.6 | 2,802 | -19.5 |
| 850 | 1,515 | 10.3 | 1,397 | -2.0 | 1,524 | -1.4 | 1,330 | -9.7 |
| 925 | -- | -- | 722 | -0.2 | 844 | 3.8 | 675 | -8.9 |





**Figure captions**

Figure 1: Maps of the deposits investigated in this work: (a) Mount St. Helens, 18 May 1980; (b) Crater Peak, 16-17 September, 1992; (c) Ruapehu, 17 June, 1996; and (d) Redoubt, 23 March, 2009. Isomass lines for Mount St. Helens were digitized from Fig. 338 in Sarna-Wojcicki et al. (1981); for Crater Peak from Fig. 15 in McGimsey et al. (2001); for Ruapehu from Fig. 1 of Bonadonna and Houghton (2005); and for Redoubt from Wallace et al. (2013). Isomass values are all in kg m$^{-2}$. Colored markers represent locations where isomass was sampled, with colors corresponding to the mass load shown in the color table. Black dashed lines indicate the dispersal axis. Sample locations for Mount St. Helens taken from supplementary material in Durant et al. (2009); for Redoubt from Wallace et al. (2013), for Crater Peak from McGimsey et al. (2001) and for Ruapehu, from data posted online at the IAVCEI Commission on Tephra Hazard Modeling database (http://dbstr.ct.ingv.it/iavcei/ (Bonadonna and Houghton, 2005; Bonadonna et al., 2005)).

Figure 2: Total particle size distribution for each of the deposits studied: (a) Mount St. Helens, (b) Crater Peak (Mount Spurr), (c) Ruapehu, and (d) Redoubt. Gray bars show the original TPSD before aggregation. Black bars show the sizes not involved in aggregation; red bars show sizes of aggregate classes used in Figs. 10-13.

Figure 3: Mass load versus downwind distance along the dispersal axis for the deposits of (a) Mount St. Helens, (b) Crater Peak (Mount Spurr), (c) Ruapehu, and (d) Redoubt. Squares indicate sample points within 20 km of the dispersal axis, with the grayscale value indicating the distance from the dispersal axis following the colorbar in (a). The dash trend lines represent interpolated values of the mass load that are compared with modeled values to calculate $\Delta^2_{downwind}$.

Figure 4: Log mass load versus the square root of the area within isomass lines mapped for the (a) Mount St. Helens; (b) Crater Peak (Spurr); (c) Ruapehu; and (d) Redoubt deposits. Also shown are best-fit lines, drawn by visual inspection, using either one line segment (Ruapehu, Redoubt) or two, where justified (Spurr, St. Helens). Triangular markers are marked with labels indicating the approximate percentage of the deposit mass lying inboard of these points, as calculated using equations derived from Fierstein and Nathenson (1992).

Figure 5: (a) Transport distance versus average fall velocity, assuming a 15.1 m s$^{-1}$ wind speed, equal to the average wind speed at Mount St. Helens between 0 and 15 km, and a fall distance



of 15 km. The vertical shaded bar represents the distance of Ritzville. Labels on dots give the
average diameter of a round aggregate having a density of 600 kg m$^{-3}$ and the given fall velocity.
(b) Average fall velocity between 0 and 15 km elevation, versus aggregate diameter, for round
aggregates having densities ranging from 200 to 2,500 kg m$^{-3}$. The horizontal shaded bar
represents the range of average fall velocities that would land in Ritzville. Fall velocities are
calculated using relations of Wilson and Huang (1979).
Figure 6: Deposit maps for simulations using a single size class representing an aggregate with
phi size 1.9 and density 600 kg m$^{-3}$, using three shape factors: (a) $F$=0.44; (b) $F$=0.7; and (c)
$F$=1.0. Inset figures illustrate ellipsoids having the given shape factor, assuming b=(a+c)/2.
Figure 7: Deposit map for simulations using a single size class representing an aggregate with
$F$=1.0, phi size 2.4$\phi$ and density 600 kg m$^{-3}$. Figs. 7a, b, and c, illustrate the deposit distribution
using Suzuki $k$ values of 4, 8, and 12, while Fig. 7d illustrates the deposit distribution resulting
from release of all the erupted mass from a single node at the top of the plume. Inset plots
schematically illustrate the vertical distribution of mass with height in the plume for each of
these cases. Simulations used other input values as given in Table 1. Colored dots represent
sample locations with colors indicating the sampled mass load, as in Fig. 1a.
Figure 8: Results of Mount St. Helens simulations using a single size class of round aggregates
in each simulation: $\phi$=1.8, 2.0, 2.2, 2.4, and 2.6 in (a), (b), (c), (d), and (e); (f) shows the mapped
mass load, digitized from Fig. 338 in Sarna-Wojcicki et al. [1981]. Markers in each figure
provide the sample locations, with colors indicating the mass load measured at each location,
as shown in the color bar. Lines are contours of mass load with colors giving their values. The
mass load values of the contour lines, from lowest to highest, are 0.01, 0.1, 0.5, 1, 5, 10, 20, 30,
50, 80, and 100 kg m$^{-2}$ respectively.
Figure 9: Contours of $\Delta^2$ (left column), $\Delta^2_{downwind}$ (middle column), and $\Delta^2_{area}$ (right column) as
a function of $\sigma_{agg}$ and $\mu_{agg}$ for deposits from Mount St. Helens (top row); Crater Peak (Mount
Spurr, second row); Ruapehu (third row), and Redoubt (bottom row). The values of these
contour lines are indicated by the color using the color bar at the right. Maximum and minimum
values in the color scale are given within each frame. The best agreement between model and
mapped data is indicated by the deep blue and purple contours; the worst is indicated by the
yellow contours. Regions of each plot where agreement is best is indicated by the word "Lo".



Figure 10: Results of the Mount St. Helens simulation that provides approximately the best fit
to mapped data ($\mu_{agg}$ =2.4ϕ and $\sigma_{agg}$ =0.3ϕ). (a) Deposit map with modeled isomass lines and
dots that represent field measurements with colors indicating the field values of the mass load,
corresponding to the color bar at left.  The black dashed line indicates the dispersal axis of the
mapped deposit whereas the solid black line with dots indicates the dispersal axis of the
modeled deposit (the latter lies mostly on top of the former and obscures it).  (b) Log of modeled
mass load versus measured mass load at sample locations.  Black dashed line is the 1:1 line;
dotted lines above and below indicate modeled values 10 and 0.1 times that measured.  Gray
dots lay outside the range of downwind distances covered by trend lines in Fig. 6 and therefore
were not included in the calculation of $\Delta^2$.  (c) Log of measured mass load (black and gray dots),
and modeled mass load (black line with dots) versus distance downwind along the dispersal
axis.  The black dashed line is the same trend line as in Fig. 7a.  Gray dots were not included in
the calculation of $\Delta^2_{area}$ .  (d) Log of mass load versus square root of area contained within
isomass lines.  Black squares are from the mapped deposit, red squares from the modeled one.
Figure 11: Results of the Crater Peak (Mount Spurr) simulation that provides approximately the
best fit to mapped data ($\mu_{agg}$ =1.8ϕ and $\sigma_{agg}$ =0.3ϕ). The features in the sub-figures are as
described in Fig. 10. "CP" in Fig. 11a refers to the Crater Peak vent.
Figure 12: Results of the Ruapehu simulation that provides approximately the best fit to mapped
data ($\mu_{agg}$ =2.4ϕ and $\sigma_{agg}$ =0.3ϕ). The features in the sub-figures are as described in Fig. 10.
Figure 13: Results of the Redoubt simulation that provides a reasonable fit to mapped data (
$\mu_{agg}$ =2.4ϕ and $\sigma_{agg}$ =0.3ϕ). The features in the sub-figures are as described in Fig. 10.
Figure 14:  Modeled mass load of the Mount St. Helens eruption for four cases using  $\mu_{agg}$
=2.4ϕ, $\sigma_{agg}$ =0.3ϕ, and different diffusion coefficients: (a) $D$=0 m$^2$ s$^{-1}$, (b) 3×10$^2$ m$^2$ s$^{-1}$, (c)
1×10$^3$ m$^2$ s$^{-1}$, and (d) 3×10$^3$ m$^2$ s$^{-1}$.  Other inputs are as given in Tables 1 and 2.  Lines are
isomass contours of modeled mass load and colored dots are sample locations.  Colors of the
dots and lines give the mass load corresponding to the color table.
Figure 15:  Modeled mass load of the Ruapehu eruption for four cases using  $\mu_{agg}$ =2.4ϕ, $\sigma_{agg}$
=0.3ϕ, and different diffusion coefficients: (a) $D$=0 m$^2$ s$^{-1}$, (b) 1×10$^2$ m$^2$ s$^{-1}$, (c) 3×10$^2$ m$^2$ s$^{-1}$,




and (d) $1 \times 10^3$ m$^2$ s$^{-1}$.  Other inputs are as given in Table 1.  Lines are isomass contours of
modeled mass load and colored dots are sample locations.  Colors of the dots and lines give the
mass load corresponding to the color table.
Figure A1: Illustration of the path taken by coarse aggregates that fallout in proximal sections,
less than a few plume heights from the source (left), and fine aggregates that fall out in distal
sections (right).  Among distal fine aggregates, we show the path taken by those that might have
formed within or below the downwind cloud as hypothesized by Durant et al. (2009) (red
dashed line), and those that were transported downwind without changing size, as calculated
by Ash3d (blue dashed line).  Also illustrated are some key processes that might influence the
distribution of fine, distal ash, including development of gravitational instability and overturn
within the downwind cloud  (Carazzo and Jellinek, 2012), and the development of
hydrometeors as descending ash approaches the freezing elevation (Durant et al., 2009).
Figures S001-S004: Figures analogous to Figs. 10, 11, 12, and 13, respectively, but with no
particle aggregation.
Figures S005-S046:  Figures analogous to Fig. 10, but for different values of $\mu_{agg}$ and $\sigma_{agg}$
given in their labels.
Figures S047-S088: Figures analogous to Fig. 11, but for different values of $\mu_{agg}$ and $\sigma_{agg}$
given in their labels.
Figures S089-S130: Figures analogous to Fig. 12, but for different values of $\mu_{agg}$ and $\sigma_{agg}$
given in their labels.
Figures S131-S172: Figures analogous to Fig. 13, but for different values of $\mu_{agg}$ and $\sigma_{agg}$
given in their labels.
Figure S173:  Figure analogous to Fig. 12, but using



Figure 1





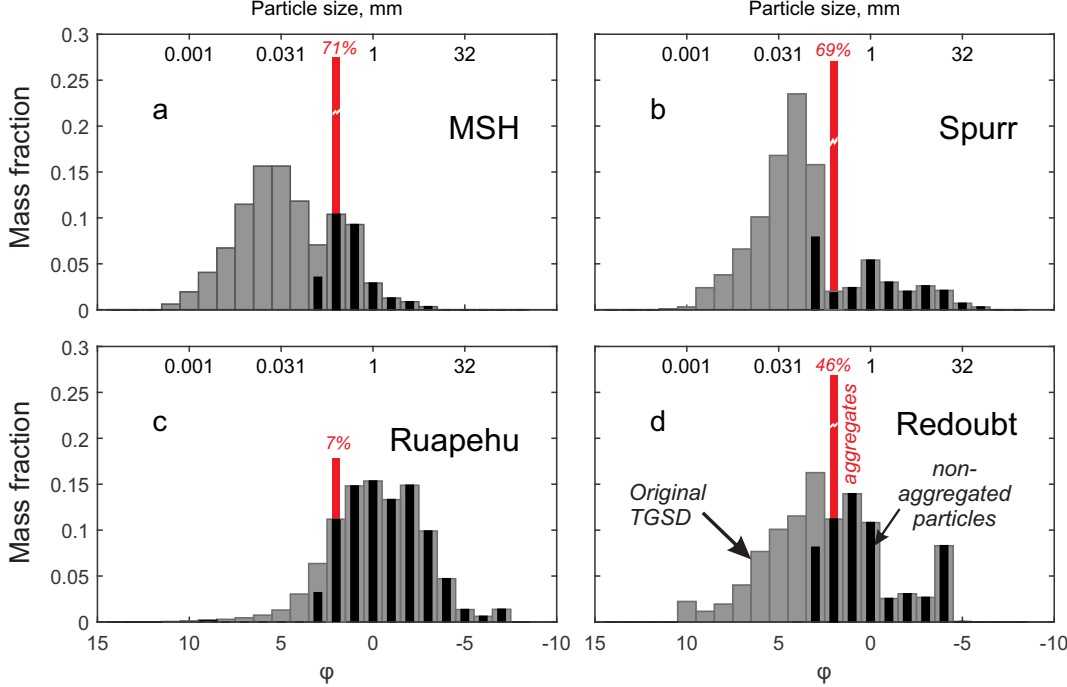

Figure 2





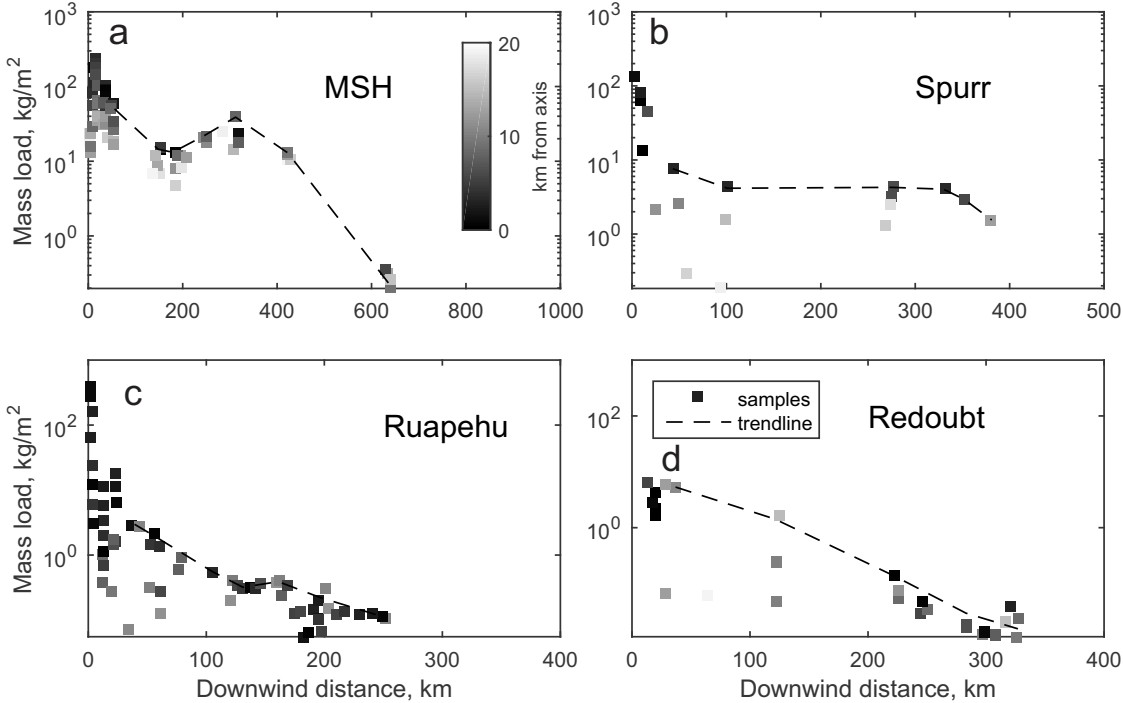

Figure 3




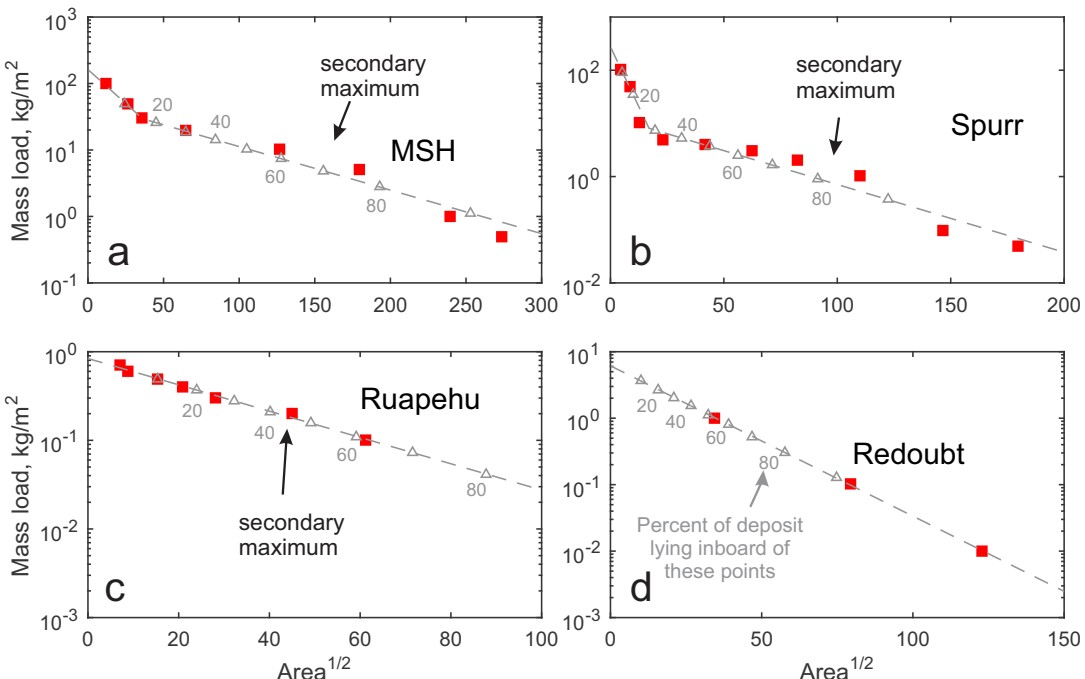

Figure 4



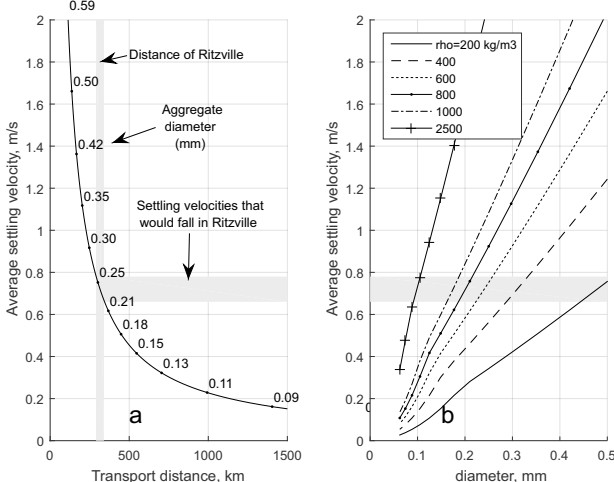

Figure 5



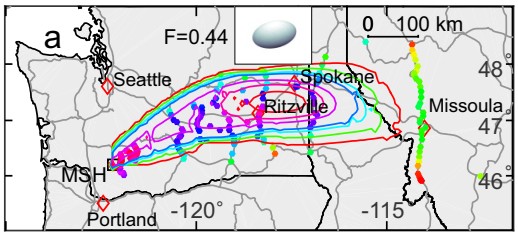

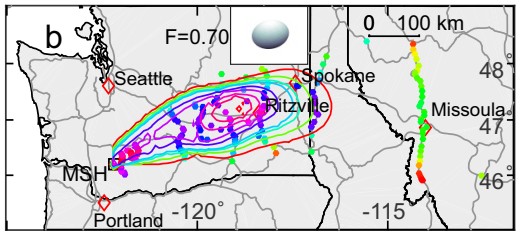

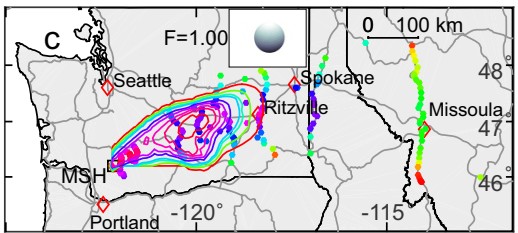

Figure 6



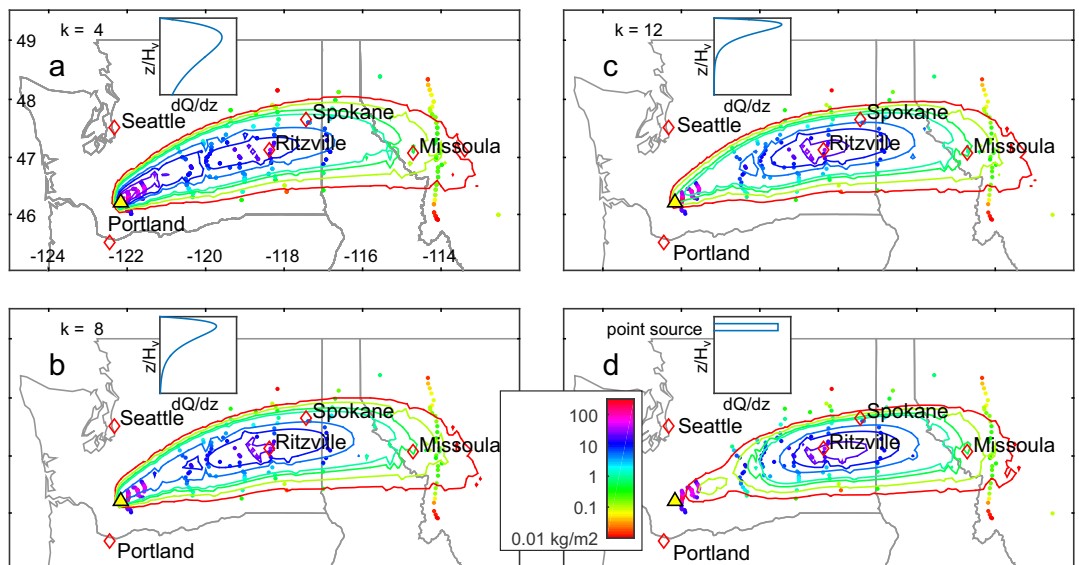



# Figure 7




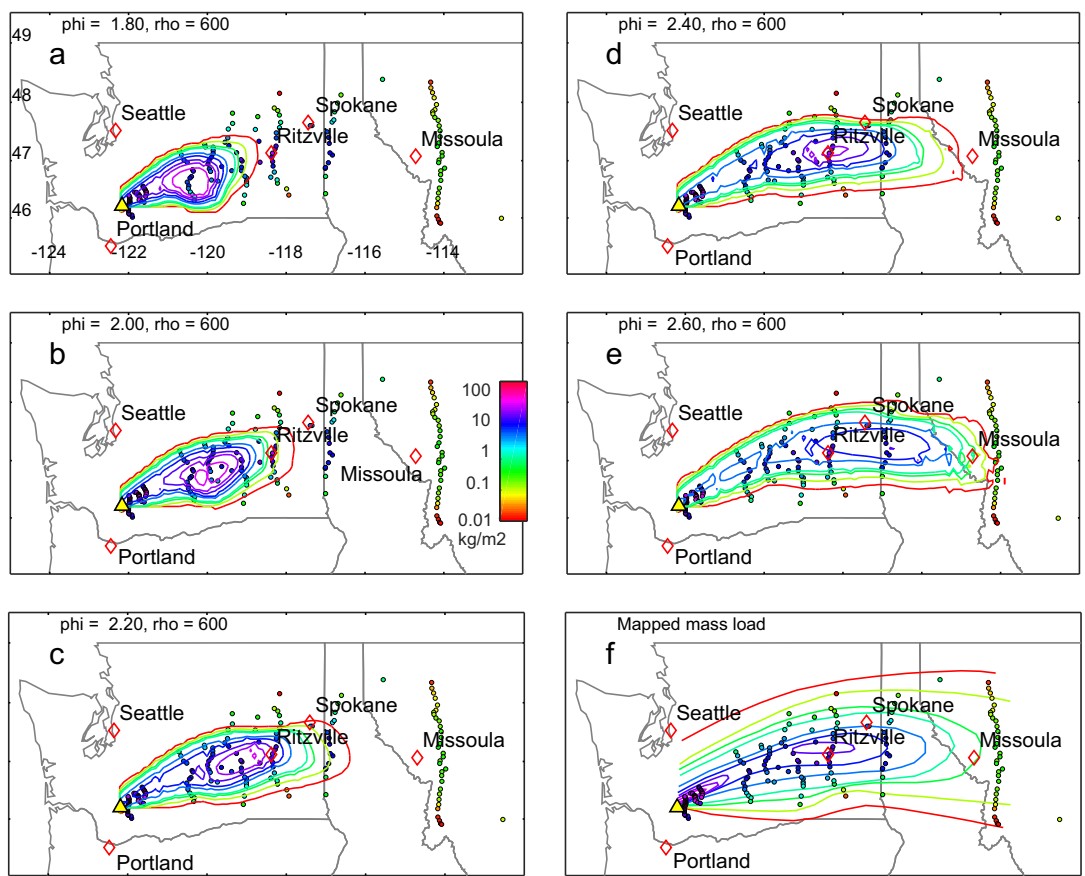

Figure 8




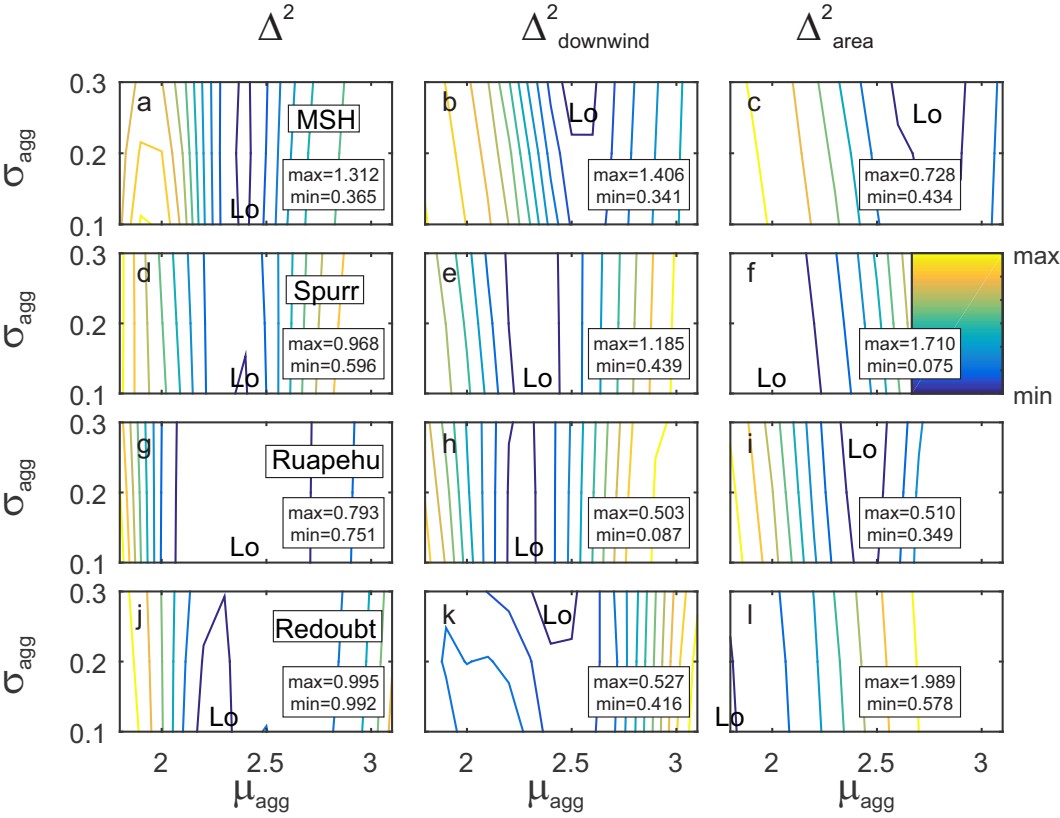

Figure 9



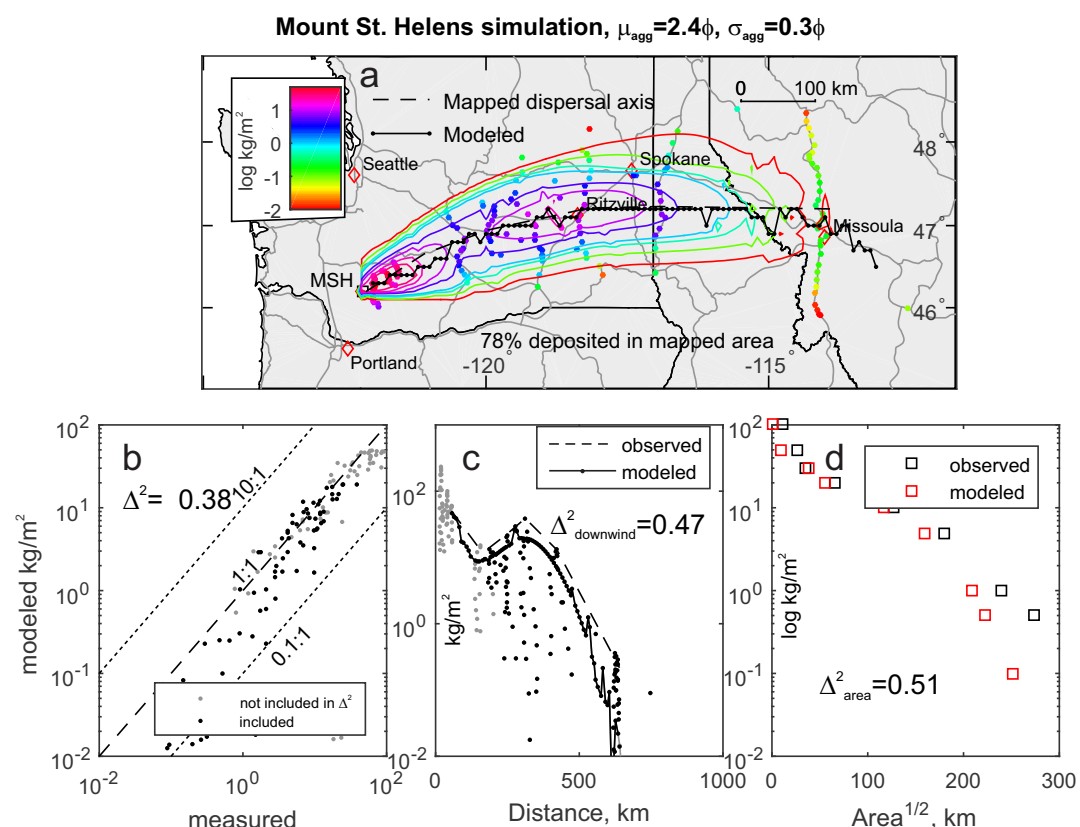

Figure 10




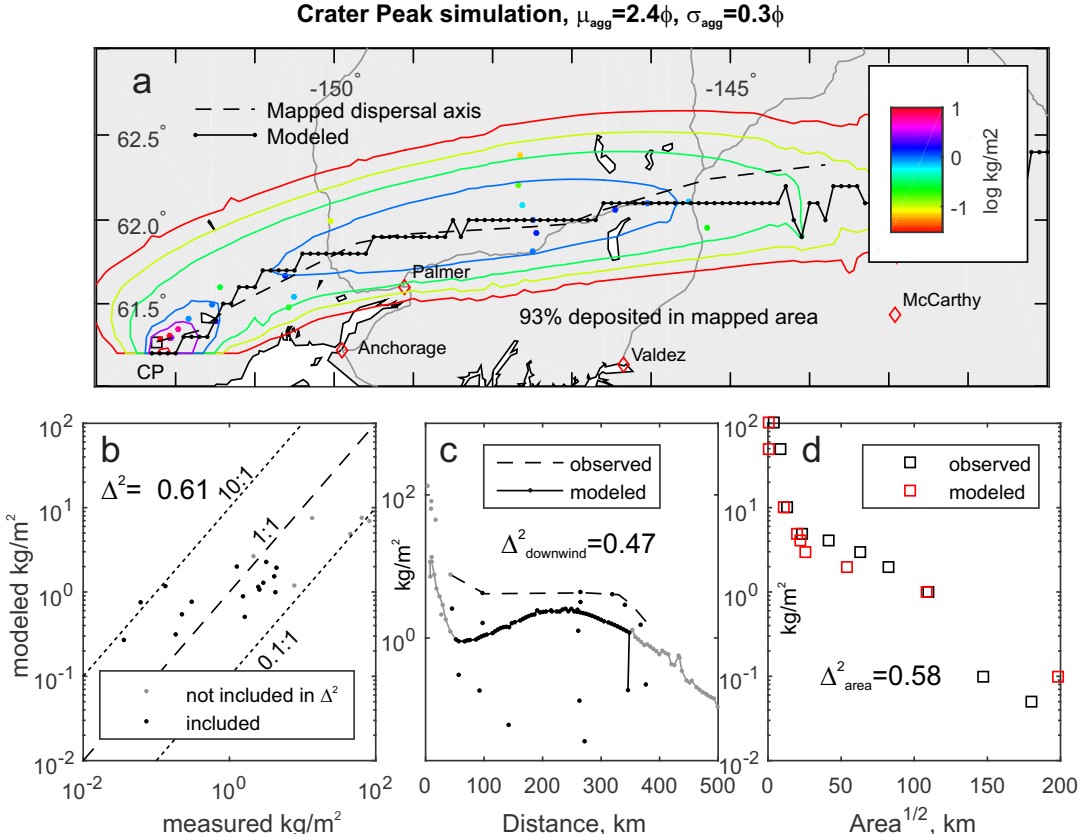

Figure 11




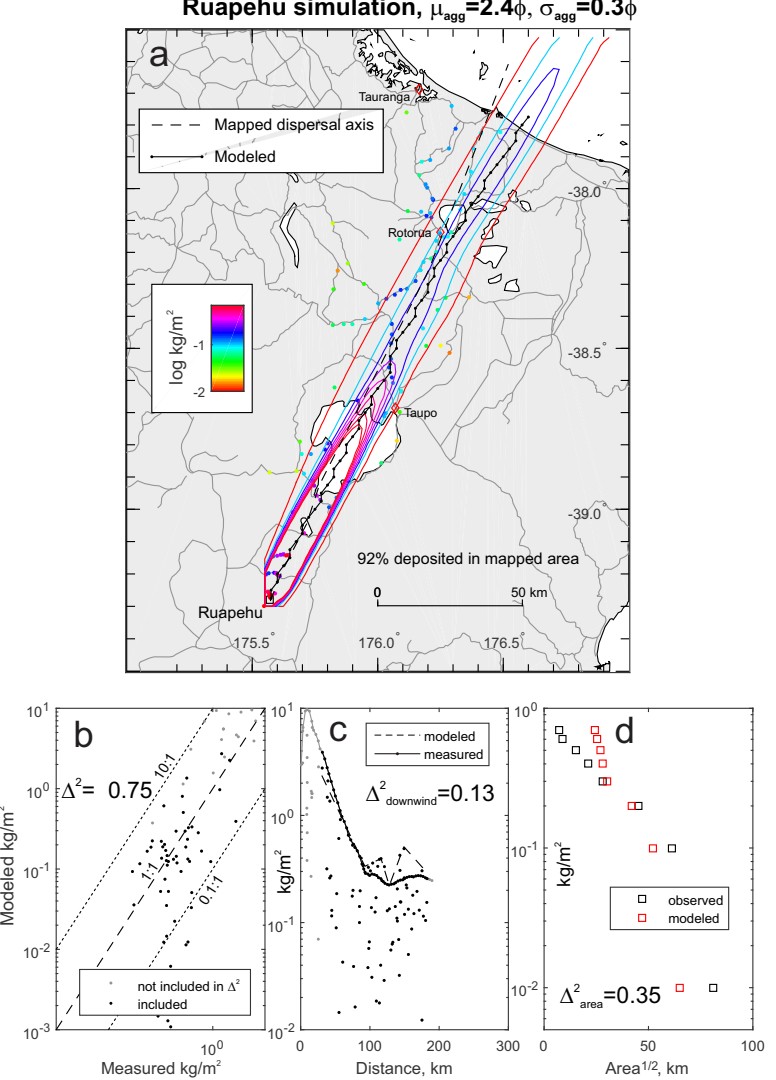

Figure 12





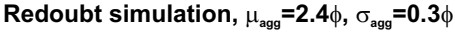

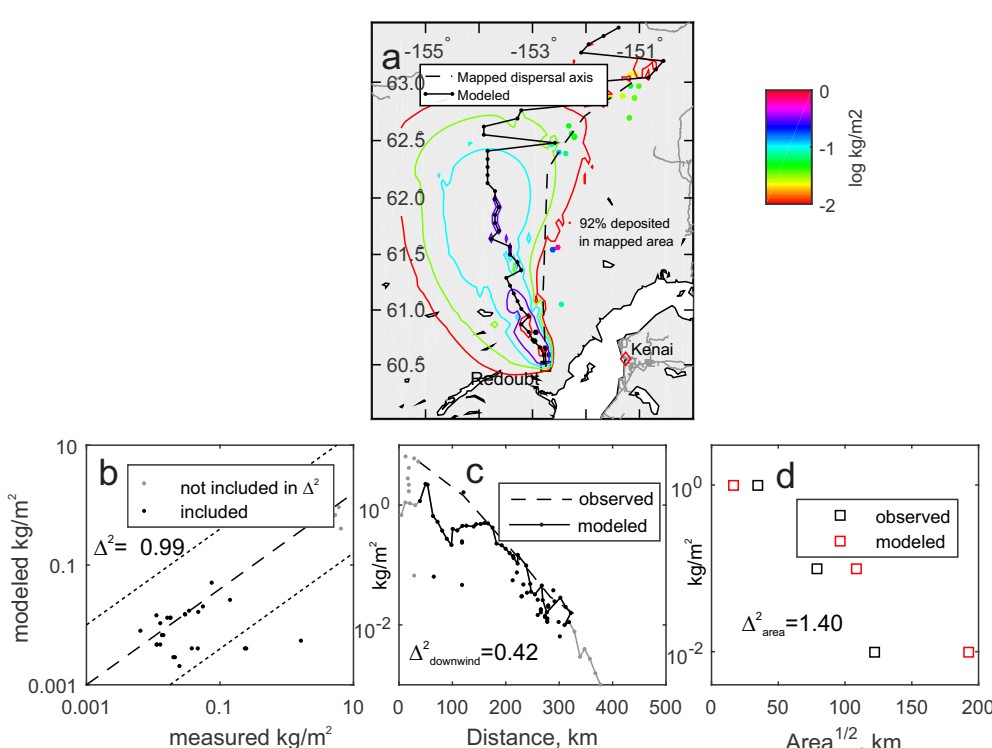

Figure 13





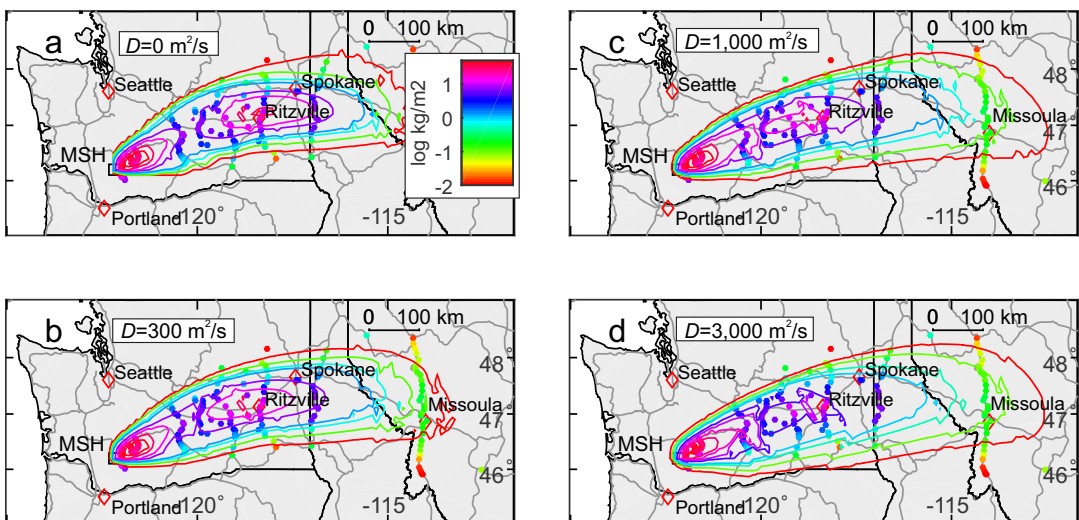

Figure 14




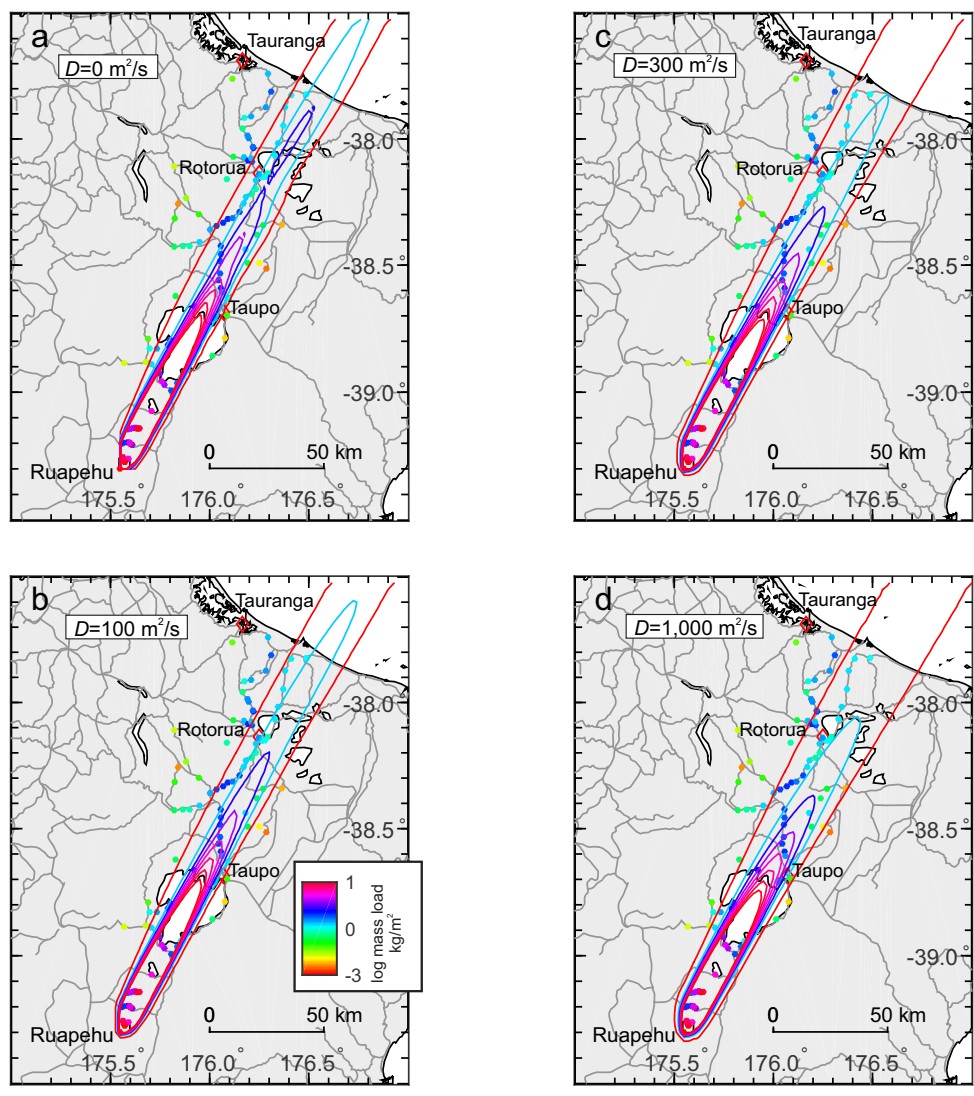

Figure 15



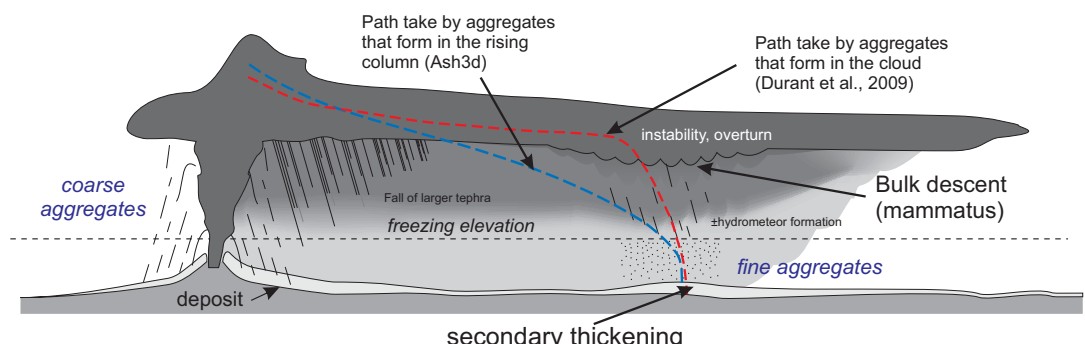

Figure A1