# Peer review of "Adjusting particle-size distributions to account for aggregation in tephra-deposit model forecasts"

_Atmospheric Chemistry and Physics, 2016_

## Referee Comment (RC1) · A. Folch (Referee) · 31 Mar 2016

A.Folch

Comments on "Adjusting particle-size distributions to account for aggregation in tephra-deposits model forecast" by Mastin et al.

This paper focus on constraining the particle size distribution (median and standard deviation for assumed Gaussian distributions in Φ) of volcanic ash aggregates. Optimal values are determined by best-fitting Ash3d model simulations with mapped deposits from 4 different well-studied events spanning over a wide range of erupted mass, intensity (column height), and granulometries. Despite the large differences in these eruptions, "model" optimal aggregate distributions are surprisingly similar, suggesting the possibility of a "cost-free" parameterization for operational forecast models that, at present, do not contemplate aggregation phenomena. This paper may not suppose a major advance from a theoretical point of view but I found it very pragmatic and very interesting for operational environments. For these reasons I do recommend its publication after addressing few minor/moderate comments detailed below.

- The parameterization scheme (section 3.2, lines 206-201) seems somehow arbitrary and could be better justified. On the other hand, if the TPSD is discretized on 1Φ intervals all the fines aggregate except for Φ=3, for which 50% of particles aggregate. This seems rather simplistic. To what extend the results depend on this choice? What if discretization is performed at 0.5Φ intervals and/or the limits are extended beyond Φ=4 (e.g. Φ=5 or 6)? Could the best-fit values (i.e. the conclusions of the paper) depend substantially on this?
- The authors find "optimal" values of aggregate density of about $\rho_{aggr}$=600 kg/m$^3$ consistent with (but denser than) previous studies and observations. It is important to mention that this is also a consequence of the settling velocity model chosen. Note that, for fine particles, the Wilson and Huang model gives smaller settling velocities than other fits (see e.g. Figure 1 in Folch 2012; Journal of Volcanology and Geothermal Research 235-236, 96–115). In other words, other velocity model using a smaller aggregate density would give exactly the same fit…

- Figure 2 is misleading because (at the beginning of the paper) gives the impression that only one aggregated bin is considered, contradicting the text. It would be much clearer if the distribution of aggregates is shown as an inset.

- Line 241-242. Values of settling velocity for a given particle strongly vary with height. Are these values at sea level or averaged?

- It is unclear to me how the modeled "dispersal axis" is obtained and why topography causes the oscillations observed in Figures 10-13.

- Line 320. Typo (And)

- Figure 10 and lines 322-327. The fact that diffusion can be ignored and still obtaining a reasonable fit is because the (Eulerian) model adds numerical diffusion. It is difficult to extract conclusions from here since this strongly depends on the numerical scheme, different from model to model.

- Line 428. "hundreds of thousands"? sure?

- Lines 424 to 444 in the discussion are rather speculative but interesting. I understand that the proposed "empirical" aggregation scheme would hold to model the finer aggregates (i.e. formed during transport), not for the larger aggregates (mm size) formed in the plume. That would explain why so different eruption conditions end up with similar mean and dispersal. Right?

---

## Referee Comment (RC2) · M. de' Michieli Vitturi (Referee) · 26 Apr 2016

**Comments on**

**"Adjusting particle-size distributions to account for  aggregation in tephra-deposit model forecasts"**

.

**by Larry G. Mastin, Alexa R. Van Eaton, and Adam J. Durant**

The paper presents a study of the modelling of volcanic ash in the atmosphere, with a particular focus on the effect of ash aggregation on depositional pattern. Several eruptions are investigated in order to find the parameters controlling aggregation, which give best fits of the deposits. To this aim, the authors employed Ash3d, an  Eulerian model that calculates tephra transport and deposition through a 3-D, time-changing  wind field.

Despite the differences in the magnitude and styles of the eruptions studied, the parameters describing ash aggregates are found to be similar for all the events.

The phenomenon investigated is interesting and very relevant for the volcanic hazard associated with ash dispersal in the atmosphere and it presents important novelties for operational model forecast. For this reason, I think that the manuscript falls into the scope of Atmospheric Chemistry and Physics and it is scientifically sufficiently sound to be published, once some points detailed below are clarified, in particular concerning the way the grain size distribution has been discretized.

- Lines 174-179. While in most of the literature the Suzuki relation is described as the distribution of mass in the column, in the original paper it is defined as "probability density diffusion". This probability is related to the mass concentration of particles leaving the column at height z in the unit time, and it is different from the concentration of particles along the column.

- Lines 189-193. In Wilson and Huang a,b and c are the principal axial lengths and not the semi-axes, and the values were measured for more than 155 particles. I am also not sure that the average value of the shape factor of 0.44 is reported in the Wilson and Huang paper.

- Section 3.2. It is not clear to me the choice of the bins for the discretization of the TPSD. Why bins of 0.5phi are used for the non-aggregated particles and bins of 0.1phi are used for the aggregated? If the settling velocity and the depositional process is sensitive to bins of 0.1phi for the aggregates, I think this should be true also for the non-aggregated particles. It is also reported that aggregates are described by a Gaussian size distribution, but the amount of fine ash assigned to different size bins, reported in Table 4, is not

representative of a Gaussian distribution. The values should be computed using the error function:

$$F(\mu + x\,\sigma) - F(\mu - x\,\sigma) = erf(x/\sqrt{2})$$

- Section 3.3. I think that the first and third indexes, defined in Table 3, should not have the square root (exponent ½).

- Section 3.4. Aggregate size. Why is the range for sigma_agg so small? Is it supported by observations or experiments? This doubt is also due to the results, showing a small sensitivity of the results with such a small range.
- Section 4.1. It is not clear why some points are excluded from the analysis in Figure 10b and 10c. In the caption it is written that for panel (b) "grey dots lay outside the range of downwind distances covered by trend lines in Fig. 6", and are excluded from the calculation of Delta^2. I don't understand why the trend lines are involved in the point-by-point index, and also why Figure 6 should be used. Also for panel (c) the caption is not clear, referring to Delta^2_area, while the figure is reporting a value for Delta^2_downwind. In any case, I think that the criteria to exclude points from the measures of the fit should be discussed more in the main text.
- Lines 322-325. It is stated that adding turbulent diffusion "visually improve the fit". For this reason, I think it would be useful to quantify how much the fit is improved, through the different statistical measures of fit presented in the paper. It is also interesting to note that the numerical results seems to show a diffusion in the results, and this is probably due to a numerical diffusion associated with the Eulerian approach. Is it possible to quantify or discuss the effects such diffusion, in relation with the grid-size?
- The choice to neglect diffusion in the model is justified by the decrease in run time from 30 to 10 minutes for operational conditions. It would be interesting to compare this time with the characteristic timing of the depositional process.

Other minor comments and typos:

   Line 258. "decreases the PERCENTAGE of erupted mass".
   Line 425. "particles from THE vent"
   Table 1: Delete UTC in start time for Spurr, Ruapehu and Redoubt. It is already reported in the field name (Start Time UTC).

M. de' Michieli Vitturi

---

## Referee Comment (RC3) · A. Neri (Referee) · 13 May 2016

Review of Mastin et al., submitted to ACP, April 2016

The manuscript aims to investigate the role and effect of particle aggregation in explosive eruptions. This is done by using a numerical model of ash dispersal and by adopting a simple parametrization of the aggregation process. Optimal parameters of such model are then derived by optimizing the comparion between model predictions and deposit evidence. The underlying hypothesis is that the effect of aggregation may be accounted for by a simple modification of the original grain-size distribution at source.

Based on the results and analysis presented the above hypothesis appears quite well justified. This is actually quite surprising given the wide range of eruptive conditions considered and the complexity of the aggregation process. On this basis, the study appears able to provide a first-order approximation of the effect of particle aggregation by simply modifying the grain-size distribution at source. This is quite relevant for improving the accuracy of operational ash disersal models.

I found the study very interesting, well-presented and certainly worth of publication after minor revision. The organization of the manuscript, as well as the figures and tables, are clear and informative. I suggest to further investigate just a few points listed below in order to make the outcomes of the study and its presentation even more robust and effective. A few minor technical points are also listed.

Main points:
- Section 3.1, lines 189-193: the Authors assume a constant particle shape factor for all particles and eruptions considered (except for the aggregates). This is probably a quite important assumption that should be acknowledged and commented given the main sensitivity of the dispersal process to such a parameter (see e.g. Scollo et al., JGR 2008; Bagheri et al., Pow. Tech. 2015; Pardini et al., JGR 2016). This is also quite evident from Fig. 6 where the shape factor of the aggregates has been varied. A similar assumption has been made for the density of the aggregates which, as explained in the text, also varies largely (lines 244-246). A brief discussion of the implications of these assumptions could be appropriate.
- Section 3.4, lines 258-261: the justification of the range of particle aggregate size and distribution (standard deviation) does not appear sufficiently clear as reported in the text. Why the assumption that most deposits fall in the region of interest is able to constrain the size of the aggregate? Is this valid/assumed just for the MSH case (Fig. 8) or for all the four eruptions? Also the extension of the mapped area is not clear. This key point should be better explain to me for both the mean and the standard deviation values. In particular the range of the standard deviation appears very narrow (i.e. 0.1-0.3) given the uncertainties involved and the results obtained, which, in some cases, indicate optimal values close to, or larger than, 0.3 (see Fig. 9).
- Section 4, lines 264-266 and Tab. 4. The way the aggregates are assigned to the various bins is not clear. In particular the distributions shown in

Tab. 4 are not Gaussian as expected. This should be corrected. It would be also interesting to see the effect of a different discretization of the Phi units of the aggreagates so to estimate the effects on the optimal parameters (units of 0.2 or 0.5 Phi instead of 0.1).

- Section 4, lines 291-297: in the description of the consistency with other studies the Authors could also mention the studies of Biass et al. (NHESS 2014) and Barsotti et al. (BV 2015) on Icelandic volcanoes and Vesuvius, respectively, that show similar optimal parameters of the aggregation process.
-

Minor technical points:
- Line 374: D should be 3x10^2.
- Line 850: Fig. 6 should be replaced by Fig. 3?
- Line 853: Fig. 7a should be replaced by Fig. 3a?

Augusto Neri

---

## Author Comment (AC1) · 18 May 2016

Responses to Folch review

*Dear Arnau:*

*Thank you for your thoughtful comments to this paper. In it, I am indicating how we are changing the manuscript in response to your critique. The revised manuscript will be posted after the Discussion period has ended.*

Reviewer comments in black. *Reponses in blue italics*

- The parameterization scheme (section 3.2, lines 206---201) seems somehow arbitrary and could be better justified. On the other hand, if the TPSD is discretized on 1Φ intervals all the fines aggregate except for Φ=3, for which 50% of particles aggregate. This seems rather simplistic. To what extend the results depend on this choice? What if discretization is performed at 0.5Φ intervals and/or the limits are extended beyond Φ=4 (e.g. Φ=5 or 6)? Could the best---fit values (i.e. the conclusions of the paper) depend substantially on this?

  *We are rewriting Section 3.2 to explain more clearly why we chose our parameterization scheme. It was based on experimental field observations of grain sizes that aggregate under different circumstances. Some explanation was already in an appendix, which has been deleted and its material moved into the main text. Figure A1 was also moved into the main paper and is now Figure 2.*

  *If we had chosen different particle-size thresholds for aggregation, the main effect would be to alter the mass that contributes to the secondary thickness maximum by several percent to tens of percent. For Mount St. Helens, about 10% of the erupted mass lies between phi=2 and phi=4. For Spurr, Ruapehu, and Redoubt, the percentages are 28%, 6% and 11%. These values reflect the variability in mass of the secondary maximum that could result from different choices of the aggregation-size threshold. In Section 3.2 we are adding a few sentences pointing this out.*

- The authors find "optimal" values of aggregate density of about $\rho_{aggr}$=600 kg/m3 consistent with (but denser than) previous studies and observations. It is important to mention that this is also a consequence of the settling velocity model chosen. Note that, for fine particles, the Wilson and Huang model gives smaller settling velocities than other fits (see e.g. Figure 1 in Folch 2012; Journal of Volcanology and Geothermal Research 235---236, 96–115). In other words, other velocity model using a smaller aggregate density would give exactly the same fit…

  *Actually, rho_agg was chosen, rather than being obtained by optimization as we did with mu_agg and sigma_agg. We chose 600 kg/m3 because it was toward the middle of the very large range of densities observed for aggregates. We are adding a paragraph to the end of Section 3.2 explaining this. We will emphasize that this choice may lead to an over- or underestimate of aggregate sizes. Our objective in this study is not to constrain the size of real aggregates, but to find a combination of parameters that can successfully replicate observed deposits. We will make this point more clearly in the first paragraph of the Discussion section.*

  *Thank you also for pointing our the dependence of our results on the chosen fall velocity. We will add a brief statement in Section 3.4 noting this.*

- Figure 2 is misleading because (at the beginning of the paper) gives the impression that only one aggregated bin is considered, contradicting the text. It would be much clearer if the distribution of aggregates is shown as an inset.

  *Yes, you're right. We will modify it to show real histogram insets.*

- Line 241---242. Values of settling velocity for a given particle strongly vary with height. Are these values at sea level or averaged?

  *As stated in the text and in the caption to Figure 5 (now 6), the fall velocities are averages. Specifically, they were averages of calculations made at 1-km intervals in the atmosphere, from 0 to 15 km. We will add this to the figure caption.*

- It is unclear to me how the modeled "dispersal axis" is obtained and why topography causes the oscillations observed in Figures 10---13.

  *The dispersal axes in Figures 10-13 are determined by finding the ground cell in each row (for Figs. 10, 11) or column (Figs. 12-13), with the highest mass load. The algorithm that finds this cell reads from an ASCII output file that give mass load at each cell center, in kg/m2, to three decimal places. At distal locations, the maximum load along a row or column may not be much greater than the precision of the output, causing a jagged appearance when spurious cells are picked. We will add an explanation to the caption of the new Fig. 11 that now explains the calculation.*

- Line 320. Typo (And)

  *Corrected—thanks.*

- Figure 1 and lines 322---327. Th fact that diffusion can be ignore and still obtaining a reasonable fit is because the (Eulerian) model adds numerical diffusion. It is difficult to extract conclusions from here since this strongly depends on the numerical scheme, different from model to model.

  *Good point. At the end of Section 4.1 we will add a sentence noting that these results may be different in other models or model configurations.*

- Line 428. "hundreds to thousands"? sure?

  *Changed to "many". (I was actually referring to the number of collisions per cubic meter per second, not per particle per second. But it wasn't clear).*

- Lines 424 to 444 in the discussion are rather speculative but interesting. I understand that the proposed "empirical" aggregation scheme would hold to model the finer aggregates (i.e. formed during transport), not for the larger aggregates (mm size) formed in the plume. That would explain why so different eruption conditions end up with similar mean and dispersal. Right?

  *If I understand your point, you seem to be suggesting that perhaps we're able to match these four deposits with similar aggregate sizes in part because we're excluding other, near-source processes, that could produce a more complicated and disparate outcome? If that's your point, I would agree. It doesn't appear that you are suggesting that anything needs to be changed in this passage; so no changes have been made.*

---

## Author Comment (AC2) · 18 May 2016

*Dear Mattia:*

*Thank you for your extensive comments to our paper. Some of your key points have prompted us to re-do all the calculations and re-formulate most of the figures. The details are below. My comments are in blue italics. Yours are in black.*

*Larry Mastin*

The paper presents a study of the modelling of volcanic ash in the atmosphere, with a particular focus on the effect of ash aggregation on depositional pattern. Several eruptions are investigated in order to find the parameters controlling aggregation, which give best fits of the deposits. To this aim, the authors employed Ash3d, an Eulerian model that calculates tephra transport and deposition through a 3-D, time-changing wind field.

Despite the differences in the magnitude and styles of the eruptions studied, the parameters describing ash aggregates are found to be similar for all the events.

The phenomenon investigated is interesting and very relevant for the volcanic hazard associated with ash dispersal in the atmosphere and it presents important novelties for operational model forecast. For this reason, I think that the manuscript falls into the scope of Atmospheric Chemistry and Physics and it is scientifically sufficiently sound to be published, once some points detailed below are clarified, in particular concerning the way the grain size distribution has been discretized.

• Lines 174-179. While in most of the literature the Suzuki relation is described as the distribution of mass in the column, in the original paper it is defined as "probability density diffusion". This probability is related to the mass concentration of particles leaving the column at height z in the unit time, and it is different from the concentration of particles along the column.

*Thank you for pointing this out. We will modify the text to indicate that we are using a modified version of the Suzuki equation, and that we are using this formula as a simple parameterization of mass distribution with height, with no attempt to relate it to physical process. The difference between a probability density function (which would not apply to our Eulerian model) and a function defining mass distribution in the column seems minor to me, unless I am misunderstanding something.*

• Lines 189-193. In Wilson and Huang a, b and c are the principal axial lengths and not the semi-axes, and the values were measured for more than 155 particles. I am also not sure that the average value of the shape factor of 0.44 is reported in the Wilson and Huang paper.

*Thank you. This was a typo, not an error in our calculations. We have corrected it in the text.*

*You're right that the average shape factor of 0.44 was not reported in Wilson and Huang. We used their data to calculate an average shape factor. We will reword the relevant sentence in section 3.1 to make this clearer.*

• Section 3.2. It is not clear to me the choice of the bins for the discretization of the TPSD. Why bins of 0.5phi are used for the non-aggregated particles and bins of 0.1phi are used for the aggregated? If the settling velocity and the depositional process is sensitive to bins of 0.1phi for the aggregates, I think this should be true also for the non-aggregated particles.

*Bins of 0.5 phi or coarser were used for the non-aggregated particles based on what was available in the published literature for these deposits. The finer, 0.1 phi bins were used for aggregates because, as shown in Figs. 5 and 8, where the aggregates land is highly sensitive to aggregate size, for the rather narrow range of sizes and densities that would put fine ash at medial distances. For non-aggregated grains, this high sensitivity is only true for particles ~50-100 microns, as illustrated in Fig. 5. Most particles of this size have already aggregated. We will add a paragraph to Section 3.4 pointing out these constraints.*

It is also reported that aggregates are described by a Gaussian size distribution, but the amount of fine ash assigned to different size bins, reported in Table 4, is not representative of a Gaussian distribution. The values should be computed using the error function:

F(mu+x sigma)-F(mu-x sigma) = erf(x/sqrt(2))

*You're totally right (gasp!); our distribution is not strictly Gaussian. And the values of sigma_agg were inaccurate for the distributions given. We are modifying the values in Table 4 and re-running all the simulations so that truly Gaussian distributions are represented. In the caption to Table 4 we will also describe exactly how these values are derived (i.e. using a Gaussian formula). This change required us to re-derive Figures 10-13, 15, 16, all the supplementary figures. It also changed the results slightly, requiring slight rewording in the Results section.*

• Section 3.3. I think that the first and third indexes, defined in Table 3, should not have the square root (exponent ½).

*Thank you. This has been changed, and the error indexes recalculated.*

• Section 3.4. Aggregate size. Why is the range for sigma_agg so small? Is it supported by observations or experiments? This doubt is also due to the results, showing a small sensitivity of the results with such a small range.

*The small range that we use is a consequence of the high sensitivity between aggregate size and distance traveled (Fig. 5). For each simulation, we wanted to use a size distribution such that the range of distances traveled between the smallest and largest aggregates was a few hundred kilometers, as illustrated in Fig. 8. This limited the range of aggregate sizes to tenths of a phi unit. Broadening the size range would have caused a large fraction of aggregates to deposit outside the range of distances we were studying. This point was made in Section 3.4.*

*However to accommodate this concern, we have slightly broadened our range of sizes by adding a size distribution to Table 4 that spans 0.8 phi units. The sigma_agg value is still small (0.3 phi), but larger than previously. When calculated properly using a Gaussian best-fit, our old maximum sigma_agg value was 0.12.*

*With this new analysis, we can show that none of the optimal fits in Fig. 9 occur at the maximum value of sigma_agg, suggesting that the range is now large enough to include the optimal value. Also, in the supplementary figures we will show that, when sigma_agg=0.3, the secondary thickness maximum is broader and less thick than observed, for example, at Mount St. Helens (Fig. S028) and Ruapehu (Fig. S128). At Spurr the secondary maximum is too poorly mapped to make this comparison, and at Redoubt no secondary maximum was mapped.*

*If we had compared the model result with more proximal sample locations, it is likely we would have obtained a wider optimal range of aggregate sizes. We chose not to include more proximal locations because the indexes we used, particularly delta^2, can be overwhelmed by proximal sample points, since their importance is directly proportional to the absolute value of the difference in mass load between the model and the measured deposit. Proximal deposition also involves processes such as hail-forming aggregation or fallout from the vertical column, that are not accurately simulated in a widespread fallout model like Ash3d. Finally, if we had included these proximal sample locations, the optimal aggregate-size distribution would probably not have produced a secondary thickness maximum. This is a key feature of three of these deposits. Not reproducing it would have yielded an unacceptable result in our opinion.*

*We are substantially revising Section 3.1 and now emphasize some of these points in that section.*

• Section 4.1. It is not clear why some points are excluded from the analysis in Figure 10b and 10c. In the caption it is written that for panel (b) "grey dots lay outside the range of downwind distances covered by trend lines in Fig. 6", and are excluded from the calculation of Delta^2. I don't understand why the trend lines are involved in the point-by-point index, and also why Figure 6 should be used.

*Values of Delta^2 can be dominated by differences in proximal locations, where mass per unit area is greatest, and where processes such as fallout from the vertical column are not accurately simulated. Therefore we exclude these proximal points from the calculation. At the beginning of Section 3.4 we noted that we ignore proximal fallout, but perhaps didn't do an adequate job explain why. We will modify the explanation of the point-by-point method in Section 3.3 to add this explanation.*

Also for panel (c) the caption is not clear, referring to Delta^2_area, while the figure is reporting a value for Delta^2_downwind.

*Thanks for pointing out this typographical error. It's now corrected.*

In any case, I think that the criteria to exclude points from the measures of the fit should be discussed more in the main text.

*I think the above-mentioned changes to Section 3.3 address this.*

• Lines 322-325. It is stated that adding turbulent diffusion "visually improve the fit". For this reason, I think it would be useful to quantify how much the fit is improved, through the different statistical measures of fit presented in the paper.

*Thanks for the suggestion. We tried this, and found that delta^2 actually shows a worse fit for the MSH case when diffusion is turned on! Apparently, the improved fit on the margins of the deposit is more than offset but poorer fit along the dispersal axis. We will note that in the new version.*

It is also interesting to note that the numerical results seems to show a diffusion in the results, and this is probably due to a numerical diffusion associated with the Eulerian approach. Is it possible to quantify or discuss the effects such diffusion, in relation with the grid-size?

*I'm not sure. At the moment, I can't think of how this would be done.*

• The choice to neglect diffusion in the model is justified by the decrease in run time from 30 to 10 minutes for operational conditions. It would be interesting to compare this time with the characteristic timing of the depositional process.

*This might be beyond the scope of the paper, but an interesting problem.*

---

## Author Response (AR1)

Dear Dr. Tesche:

Below you will find the colleague reviews to this paper, along with our response. Reviewer comments are in black. Responses in *blue italics*.

Appended below the comments and responses is a copy of the manuscript text with changes tracked. The revised manuscript with figures is provided as a separate file.

We hope you find these responses adequate to merit publication.

Sincerely,

Larry Mastin

**Folch Review**

 The parameterization scheme (section 3.2, lines 206---201) seems somehow arbitrary and could be better justified. On the other hand, if the TPSD is discretized on 1Φ intervals all the fines aggregate except for Φ=3, for which 50% of particles aggregate. This seems rather simplistic. To what extend the results depend on this choice? What if discretization is performed at 0.5Φ intervals and/or the limits are extended beyond Φ=4 (e.g. Φ=5 or 6)? Could the best---fit values (i.e. the conclusions of the paper) depend substantially on this?

We have rewritten Section 3.2 to explain more clearly why we chose our parameterization scheme. It was based on experimental field observations of grain sizes that aggregate under different circumstances. Some explanation was already in an appendix, which has been deleted and its material moved into the main text. Figure A1 was also moved into the main paper and is now Figure 2.

In Section 3.2 (lines 250-255) we added a few sentences indicating the effect on results of different choices of the aggregation-size threshold. The main effect is to alter the mass that contributes to the secondary thickness maximum by several percent to tens of percent.

 The authors find "optimal" values of aggregate density of about paggr=600 kg/m3 consistent with (but denser than) previous studies and observations. It is important to mention that this is also a consequence of the settling velocity model chosen. Note that, for fine particles, the Wilson and Huang model gives smaller settling velocities than other fits (see e.g. Figure 1 in Folch 2012; Journal of Volcanology and Geothermal Research 235---236, 96–115). In other words, other velocity model using a smaller aggregate density would give exactly the same fit...

Actually, rho\_agg was chosen, rather than being obtained by optimization as we did with mu\_agg and sigma\_agg. We chose 600 kg/m3 because it was toward the middle of the very large range of densities observed for aggregates. We have added a paragraph to the end of Section 3.2 explaining this. We also now emphasize that this choice may lead to an over- or underestimate of aggregate sizes. Our objective in this study is not to constrain the size of real aggregates, but to find a combination of parameters that can successfully replicate observed deposits. We now make this point in the first paragraph of the Discussion section.

In Section 3.4 we also added a brief section point out that our results are dependent on the fall model chosen.

• Figure 2 is misleading because (at the beginning of the paper) gives the impression that only one aggregated bin is considered, contradicting the text. It would be much clearer if the distribution of aggregates is shown as an inset.

Yes, you're right. We have modified Figure 2 (now Figure 3) to show inset histograms of aggregate sizes.

• Line 241---242. Values of settling velocity for a given particle strongly vary with height. Are these values at sea level or averaged?

As stated in the text and in the caption to Figure 5 (now 6), the fall velocities are averages. Specifically, they were averages of calculations made at 1-km intervals in the atmosphere, from 0 to 15 km. We now say this in the figure caption.

• It is unclear to me how the modeled "dispersal axis" is obtained and why topography causes the oscillations observed in Figures 10---13.

The dispersal axes in Figures 10-13 (now 11-14) are determined by finding the ground cell in each row (for Figs. 10, 11) or column (Figs. 12-13), with the highest mass load. The algorithm that finds this cell reads from an ASCII output file that give mass load at each cell center, in kg/m2, to three decimal places. At distal locations, the maximum load along a row or column may not be much greater than the precision of the output, causing a jagged appearance when spurious cells are picked. We have added an explanation to the caption of the new Fig. 11 that now explains the calculation.

• Line 320. Typo (And)

**Corrected—thanks.**

• Figure 1 and lines 322---327. The fact that diffusion can be ignore and still obtaining a reasonable fit is because the (Eulerian) model adds numerical diffusion. It is difficult to extract conclusions from here since this strongly depends on the numerical scheme, different from model to model.

Good point. At the end of Section 4.1 we have added a sentence noting that these results may be different in other models or model configurations.

• Line 428. "hundreds to thousands"? sure?

Changed to "many"

• Lines 424 to 444 in the discussion are rather speculative but interesting. I understand that the proposed "empirical" aggregation scheme would hold to model the finer aggregates (i.e. formed during transport), not for the larger aggregates (mm size) formed in the plume. That would explain why so different eruption conditions end up with similar mean and dispersal. Right?

If I understand your point, you seem to be suggesting that perhaps we're able to match these four deposits with similar aggregate sizes in part because we're excluding other, near-source processes, that could produce a more complicated and disparate outcome. If that's your point, I agree. It doesn't appear that you are suggesting that anything needs to be changed in this passage.

**di'Michieli Vitturi Review**

The paper presents a study of the modelling of volcanic ash in the atmosphere, with a particular focus on the effect of ash aggregation on depositional pattern. Several eruptions are investigated in order to find the parameters controlling aggregation, which give best fits of the deposits. To this aim, the authors employed Ash3d, an Eulerian model that calculates tephra transport and deposition through a 3-D, timechanging wind field.

Despite the differences in the magnitude and styles of the eruptions studied, the parameters describing ash aggregates are found to be similar for all the events.

The phenomenon investigated is interesting and very relevant for the volcanic hazard associated with ash dispersal in the atmosphere and it presents important novelties for operational model forecast. For this reason, I think that the manuscript falls into the scope of Atmospheric Chemistry and Physics and it is scientifically sufficiently sound to be published, once some points detailed below are clarified, in particular concerning the way the grain size distribution has been discretized.

• Lines 174-179. While in most of the literature the Suzuki relation is described as the distribution of mass in the column, in the original paper it is defined as "probability density diffusion". This probability is related to the mass concentration of particles leaving the column at height z in the unit time, and it is different from the concentration of particles along the column.

Thank you for pointing this out. We have modified the text to indicate that we are using a modified version of the Suzuki equation, and that we are using this formula as a simple parameterization of mass distribution with height, with no attempt to relate it to physical process. The difference between a probability density function (which would not apply to our Eulerian model) and a function defining mass distribution in the column seems minor to me, unless I am misunderstanding something.

• Lines 189-193. In Wilson and Huang a, b and c are the principal axial lengths and not the semi-axes, and the values were measured for more than 155 particles. I am also not sure that the average value of the shape factor of 0.44 is reported in the Wilson and Huang paper.

*Thank you.* This was a typo, not an error in our calculations. We have corrected it in the text (3rd paragraph of Section 3.1).

You're right that the average shape factor of 0.44 was not reported in Wilson and Huang. We used their data to calculate an average shape factor. We have reworded the last sentence in the penultimate paragraph of section 3.1 to make this clearer.

• Section 3.2. It is not clear to me the choice of the bins for the discretization of the TPSD. Why bins of 0.5phi are used for the non-aggregated particles and bins of 0.1phi are used for the aggregated? If the settling velocity and the depositional process is sensitive to bins of 0.1phi for the aggregates, I think this should be true also for the non-aggregated particles.

Bins of 0.5 phi or coarser were used for the non-aggregated particles based on what was available in the published literature for these deposits. The finer, 0.1 phi bins were used for aggregates because, as shown in Figs. 6 and 9, where the aggregates land is highly sensitive to aggregate size, for the rather

narrow range of sizes and densities that would put fine ash at medial distances. For non-aggregated grains, this high sensitivity is only true for particles ~50-100 microns, as illustrated in Fig. 6. Most particles of this size have already aggregated. We have added a paragraph to Section 3.4 pointing out these constraints.

It is also reported that aggregates are described by a Gaussian size distribution, but the amount of fine ash assigned to different size bins, reported in Table 4, is not representative of a Gaussian distribution. The values should be computed using the error function:

**F(mu+x sigma)-F(mu-x sigma) = erf(x/sqrt(2))**

You're totally right (gasp!); our distribution is not strictly Gaussian. And the values of sigma\_agg were inaccurate for the distributions given. We have modified the values in Table 4 and re-run all the simulations so that truly Gaussian distributions are represented. In the caption to Table 4 we also describe exactly how these values are derived (i.e. using a Gaussian formula). This change required us to re-derive Figures 10-13, 15, 16, all the supplementary figures. It also changed the results slightly, requiring slight rewording in the Results section.

• Section 3.3. I think that the first and third indexes, defined in Table 3, should not have the square root (exponent ½).

**Thank you. This has been changed, and the error indexes recalculated.**

• Section 3.4. Aggregate size. Why is the range for sigma\_agg so small? Is it supported by observations or experiments? This doubt is also due to the results, showing a small sensitivity of the results with such a small range.

The small range that we use is a consequence of the high sensitivity between aggregate size and distance traveled (Fig. 6). For each simulation, we wanted to use a size distribution such that the range of distances traveled between the smallest and largest aggregates was a few hundred kilometers, as illustrated in Fig. 9. This limited the range of aggregate sizes to tenths of a phi unit. Broadening the size range would have caused a large fraction of aggregates to deposit outside the range of distances we were studying. This point was made in Section 3.4.

However to accommodate this concern, we have slightly broadened our range of sigma\_agg values. The sigma\_agg value is still small (0.3 phi), but larger than previously. When calculated properly using a Gaussian best-fit, our old maximum sigma\_agg value was 0.12.

With this new analysis, we can show that almost none of the optimal fits in Fig. 10 occur at the maximum value of sigma\_agg, suggesting that the range is now large enough to include the optimal value. Also, a perusal of the supplementary figures shows that, when sigma\_agg=0.3, the secondary thickness maximum is broader and less thick than observed, for example, at Mount St. Helens (Fig. S028) and Ruapehu (Fig. S128).

If we had compared the model result with more proximal sample locations, it is likely we would have obtained a wider optimal range of aggregate sizes. We chose not to include more proximal locations because the indexes we used, particularly delta^2, can be overwhelmed by proximal sample points, since their importance is directly proportional to the absolute value of the difference in mass load between the model and the measured deposit. Proximal deposition also involves processes such as hail-forming

aggregation or fallout from the vertical column, that are not accurately simulated in a widespread fallout model like Ash3d. Finally, if we had included these proximal sample locations, the optimal aggregate-size distribution would probably not have produced a secondary thickness maximum, because it would have been optimizing to fit the proximal deposit. The secondary thickening is a key feature of three of these deposits. Not reproducing it would have yielded an unacceptable result in our opinion.

**We have substantially revised Section 3.1 and now emphasize these points in that section.**

• Section 4.1. It is not clear why some points are excluded from the analysis in Figure 10b and 10c. In the caption it is written that for panel (b) "grey dots lay outside the range of downwind distances covered by trend lines in Fig. 6", and are excluded from the calculation of Delta^2. I don't understand why the trend lines are involved in the point-by-point index, and also why Figure 6 should be used.

Values of Delta2 can be dominated by differences in proximal locations, where mass per unit area is greatest, and where processes such as fallout from the vertical column are not accurately simulated. Therefore we exclude these proximal points from the calculation. At the beginning of Section 3.4 we note that we ignore proximal fallout, but perhaps didn't do an adequate job explain why. We have modified the explanation of the point-by-point method in Section 3.3 to add this explanation.

Also for panel (c) the caption is not clear, referring to Delta2\_area, while the figure is reporting a value for Delta2\_downwind.

**Thanks for pointing out this typographical error. It's now corrected.**

In any case, I think that the criteria to exclude points from the measures of the fit should be discussed more in the main text.

**I think the above-mentioned changes to Section 3.3 address this.**

• Lines 322-325. It is stated that adding turbulent diffusion "visually improve the fit". For this reason, I think it would be useful to quantify how much the fit is improved, through the different statistical measures of fit presented in the paper.

We tried this, and found that delta^2 actually shows a worse fit for the MSH case when diffusion is turned on! Apparently, the improved fit on the margins of the deposit is more than offset but poorer fit along the dispersal axis. We will note that in the last paragraph of section 4.1.

It is also interesting to note that the numerical results seems to show a diffusion in the results, and this is probably due to a numerical diffusion associated with the Eulerian approach. Is it possible to quantify or discuss the effects such diffusion, in relation with the grid-size?

**I'm not sure. At the moment, I can't think of how this would be done.**

• The choice to neglect diffusion in the model is justified by the decrease in run time from 30 to 10 minutes for operational conditions. It would be interesting to compare this time with the characteristic timing of the depositional process.

This might be beyond the scope of the paper, but an interesting problem.

**Neri Review**

The manuscript aims to investigate the role and effect of particle aggregation in explosive eruptions. This is done by using a numerical model of ash dispersal and by adopting a simple parametrization of the aggregation process. Optimal parameters of such model are then derived by optimizing the comparison between model predictions and deposit evidence. The underlying hypothesis is that the effect of aggregation may be accounted for by a simple modification of the original grain---size distribution at source.

Based on the results and analysis presented the above hypothesis appears quite well justified. This is actually quite surprising given the wide range of eruptive conditions considered and the complexity of the aggregation process. On this basis, the study appears able to provide a first---order approximation of the effect of particle aggregation by simply modifying the grain---size distribution at source. This is quite relevant for improving the accuracy of operational ash dispersal models.

I found the study very interesting, well---presented and certainly worth of publication after minor revision. The organization of the manuscript, as well as the figures and tables, are clear and informative. I suggest to further investigate just a few points listed below in order to make the outcomes of the study and its presentation even more robust and effective. A few minor technical points are also listed.

**Main points:**

Section 3.1, lines 189---193: the Authors assume a constant particle shape factor for all particles and eruptions considered (except for the aggregates). This is probably a quite important assumption that should be acknowledged and commented given the main sensitivity of the dispersal process to such a parameter (see e.g. Scollo et al., JGR 2008; Bagheri et al., Pow. Tech. 2015; Pardini et al., JGR 2016). This is also quite evident from Fig. 6 where the shape factor of the aggregates has been varied. A similar assumption has been made for the density of the aggregates which, as explained in the text, also varies largely (lines 244---246). A brief discussion of the implications of these assumptions could be appropriate.

Thank you for this observation. A similar point was raised in A. Folch's review, and it shows we need to emphasize that our objective is to see whether "standard" values of these parameters (even if locally unrealistic) can successfully match observations. During an eruption, these values cannot be scrutinized and there is a need to have a set of standard values that are known to work well in reproducing observations. We have added a short paragraph to the end of section 3.4 emphasizing this point, and also a couple of sentences to the first paragraph of the Discussion section. Also in the first paragraph of the Discussion section, we emphasize that our results depend on the specific inputs chosen.

 Section 3.4, lines 258---261: the justification of the range of particle aggregate size and distribution (standard deviation) does not appear sufficiently clear as reported in the text. Why the assumption that most deposits fall in the region of interest is able to constrain the size of the aggregate? We assume that most aggregates fall in the region of interest because studies suggest that most erupted mass, probably >90%, falls to form a recognizable deposit rather than transporting farther downwind as a distal cloud. This is now mentioned in Section 3.4, when describing constraints on aggregate size

Is this valid/assumed just for the MSH case (Fig. 8) or for all the four eruptions?

We use the observations from Mount St. Helens to derive these constraints, but assume it applies to all four eruptions. We now state this explicitly in Section 3.4, when describing constraints on aggregate size.

Also the extension of the mapped area is not clear.

It is the area shown in the new Fig. 9. This is now mentioned in Section 3.4, in the paragraph describing aggregate size.

This key point should be better explain to me for both the mean and the standard deviation values. In particular the range of the standard deviation appears very narrow (i.e. 0.1---0.3) given the uncertainties involved and the results obtained, which, in some cases, indicate optimal values close to, or larger than, 0.3 (see Fig. 9).

In fact, the standard deviation of aggregate sizes was even narrower than we stated. After calculating a proper Gaussian best-fit using the size distribution of our previous manuscript, the maximum value of sigma\_agg was only 0.12, not 0.3. We have revised the aggregate-size distributions as shown in Table 4, using 4 values now (sigma\_agg=0, 0.1, 0.2, 0.3). We don't think a wider distribution is justified, for two reasons

(1) using sigma\_agg=0.3, too much fine ash flows out of the model domain. For example, in the Mount St. Helens case, even for the optimal value of mu\_agg (2.4), only 75% of the erupted deposit lands within the mapped area. For mu\_agg=2.5, 2.7, and 2.9, the values drop to 70%, 60%, and 48%. These values are too low to be realistic, in our opinion.

(2) For the Mount St. Helens case, the value of sigma\_agg=0.3 produces a secondary thickening that is broader and more diffuse than observed, for example, in fig. 11c. For the other deposits, the secondary thickening is not sufficiently well defined to judge. We make this point in the first paragraph of Section 4.1.

• Section 4, lines 264---266 and Tab. 4. The way the aggregates are assigned to the various bins is not clear. In particular the distributions shown in Tab. 4 are not Gaussian as expected. This should be corrected.

This point was also made in Mattia di'Michieli Vitturi's review. We have changed the distribution of aggregates in Table 4 so that they are now Gaussian, and added an explanation of how they were calculated to the Table 4 caption. This change required us to run all the simulations again and derive new results.

It would be also interesting to see the effect of a different discretization of the Phi units of the aggreagates so to estimate the effects on the optimal parameters (units of 0.2 or 0.5 Phi instead of 0.1).

I think this point is addressed in our response to the previous bullet.

Section 4, lines 291---297: in the description of the consistency with other studies the Authors could also mention the studies of Biass et al. (NHESS 2014) and Barsotti et al. (BV 2015) on Icelandic volcanoes and Vesuvius, respectively, that show similar optimal parameters of the aggregation process.

Thanks for reminding me of these papers. I now cite them in the first paragraph of the Discussion section. It's interesting that they use a wider range of aggregate sizes, based partly on observations by Bonadonna et al. (2002) at Montserrat. We found a smaller size range to be optimal in this study. But the difference may lie partly in the fact that we optimized the fit to sample locations at distances of several tens to hundreds of kilometers. Including more proximal sample points may have resulted in a wider optimal aggregate-size range. I now point that out when citing them.

**Minor technical points:**

- Line 374: D should be 3x10^2. *Corrected—thanks!*
- Line 850: Fig. 6 should be replaced by Fig. 3? *Yes, thanks. Should now be Fig. 4.*
- Line 853: Fig. 7a should be replaced by Fig. 3a? *Yes, corrected. (now Fig. 4a)*

Augusto Neri

[revised manuscript text omitted]

| 238 | and distal, dry aggregates (Sorem, 1982). The latter type deposited over a larger area, involved               |   |
|-----|----------------------------------------------------------------------------------------------------------------|---|
| 239 | a greater fraction of the total erupted mass, and affected a greater population. Thus it is the                |   |
| 240 | process whose deposits we wish to reproduce.                                                                   |   |
| 241 | Aggregation is also a highly size-selective process. The threshold size below which most                       |   |
| 242 | particles aggregate and above which they don't varies with moisture and electrical charge,                     |   |
| 243 | ranging from several tens of microns under dry conditions, to hundreds of microns when liquid                  |   |
| 244 | water is present (Gilbert and Lane, 1994; Schumacher and Schmincke, 1995; Van Eaton et al.,                    |   |
| 245 | $\underline{2012}$ ). Our aggregation scheme is too crude to distinguish the threshold size as a function of   |   |
| 246 | atmospheric conditions, hence we use a broad range such that:                                                  |   |
| 247 | $\underbrace{For \ \phi \ge = 4, all \ ash \ aggregates}_{==}$                                                 |   |
| 248 | For $\phi \leq =2$ , no ash aggregates.                                                                        |   |
| 249 | For 4> $\phi$ >2, the mass fraction that aggregates varies linearly with $\phi$ from 1 (when $\phi = 4$ ) to 0 |   |
| 250 | (when $\phi=2$ ).                                                                                              |   |
| 251 | The TPSD used to model these four eruptions are listed in Table S1 and illustrated as gray bars                |   |
| 252 | in Fig. 3. Particle sizes that do not aggregate according to this scheme are illustrated as black              |   |
| 253 | bars. We assume that the aggregates collect into clusters having a Gaussian size distribution of               |   |
| 254 | mean $\mu_{agg}$ , and standard deviation $\sigma_{agg}$ (insets, Fig. 3). For deposit modeling, we ignore the |   |
| 255 | small fraction of the erupted mass that goes into the distal cloud, typically a few percent (Dacre             | C |
| 256 | et al., 2011; Devenish et al., 2012).                                                                          |   |
| 257 | In our study, the aggregated ash mostly deposits as a secondary thickness maximum. Different                   |   |
| 258 | choices of a threshold size for particle aggregation would influence the mass building the                     |   |
| 259 | secondary maximum. For Mount St. Helens, about 10% of the erupted mass lies between $\phi=2$                   |   |
| 260 | and $\phi$ =4. For Spurr, Ruapehu, and Redoubt, the percentages are 28%, 6% and 11%. These                     |   |
| 261 | values reflect the variability in mass of the secondary maximum that could result from different               |   |
| 262 | choices of the aggregation-size threshold.                                                                     |   |
| 263 | Aggregate Density: Liquid water also Different processes influences aggregate morphology,                      |   |
| 264 | density <del>, and rate of formation. Laboratory experiments have shown that wWet ash (>10-15</del>         |   |
| 265 | wt.% liquid water) rapidly produces dense, sub-spherical pellets with density >1,000 kg m -3        |   |

266 (Schumacher and Schmincke, 1991; Van Eaton et al., 2012):- whereas-drier conditions lead to

Field Code Changed Field Code Changed

| 267 | low-density, electrostatically-bound clusters (Schumacher and Schmincke, 1995; Van Eaton et                                      |   |
|-----|----------------------------------------------------------------------------------------------------------------------------------|---|
| 268 | al., 2012) with density in the hundreds of kilograms per cubic meter (James et al., 2002;                                        |   |
| 269 | Taddeucci et al., 2011) (Schumacher and Schmincke, 1995; James et al., 2002; Van Eaton et                                 |   |
| 270 | al., 2012). (2002) Taddeucci et al. (2011) estimated densities of dominantly several                                             | _ |
| 271 | hundred ranging from $<100$ to $>1,000$ kg m -3 in dry aggregates photographed falling 7 km from                      |   |
| 272 | the Eyjafjallajökull vent. James et al., (2003) however estimate dry aggregate densities less                                    |   |
| 273 | than 200 kg m -3 . For simplicity, we hold $\rho_{agg}$ constant at 600 kg m -3 , toward the middle of the | _ |
| 274 | observed range but higher than that of some dry aggregates. Our results, in terms of oOptimal                                    |   |
| 275 | aggregate sizes that we derive later in this paper, are determined by this assumed density, and                                  |   |
| 276 | may be larger or smaller than actual aggregate sizes depending on the density used here.                                         |   |
| 277 | The TPSD used to model these four eruptions are listed in Table S1 and illustrated in Fig. 2.                                    |   |
| 278 | We aim to adjust the TPSD in our model to better match the mapped deposits. In doing so, we                                      |   |
| 279 | assume that some fraction $(m_{agg})$ of ash smaller than some size $\phi_p^{max}$ collects into clusters having                 |   |
| 280 | a density $ ho_{agg}$ and Gaussian size distribution of mean $\mu_{agg}$ , and standard deviation $\sigma_{agg}$ . For           |   |
| 281 | deposit modeling, we ignore the small fraction of the erupted mass that goes into the distal                                     |   |
| 282 | cloud, typically a few percent (Dacre et al., 2011; Devenish et al., 2012). In the Appendix we                                   |   |
| 283 | briefly review aggregation processes. We offer the following parameterization scheme:                                            |   |
| 284 | For $\phi >=4$ , all ash aggregates                                                                                              |   |
| 285 | For $\phi \ll 2$ , no ash aggregates.                                                                                            |   |
| 286 | For $4 \ge \phi \ge 2$ , the mass fraction that aggregates varies linearly with $\phi$ from 1 (when $\phi = 4$ ) to 0            |   |
| 287 | (when $\phi=2$ ). Based on this scheme, particle sizes that aggregate are depicted as gray bars in Fig.                          |   |
| 288 | <del>2.</del>                                                                                                                    |   |
| •   |                                                                                                                                  |   |

**289 **3.3 Statistical measures of fit**

For each eruption, we have done a series of model simulations, first using the TPSD without considering aggregation, and then systematically varying  $\sigma_{agg}$  and  $\mu_{agg}$  to include the effects of aggregation. We compare the resulting deposit with the mapped deposit using three methods presented in Table 3. Each has advantages and disadvantages.

[revised manuscript text omitted]

Table 4: percentage of fine ash assigned to different size bins for different values of  $\sigma_{agg}$ . The mass fraction  $m_{\phi}$  in each bin ( $\phi$ ) was calculated using the equation for a Poisson distribution,  $m_{\phi} = (1/\sqrt{2\pi}) \exp\left\{\left[-(\phi - \mu_{agg})\right]^2/(2\sigma_{agg})^2\right\}$ . Values of  $m_{\phi}$  were then adjusted proportionally so that their sum added to 1.  $\sigma_{agg}$  -0.4 $\phi$  -0.3 $\phi$  -0.2 $\phi$  -0.1 $\phi$   $\mu_{agg}$  +0.1 $\phi$  +0.2 $\phi$  +0.3 $\phi$  +0.4 $\phi$

| $\sigma_{agg}$               | - <del>0.4</del> ¢       | -0            | <del>.3</del> 0 | <del>-0.2</del> ¢ | <del>-0.1ø</del> | $\mu_{agg}$     | <del>+0.1ø</del> | <del>+0.2\$</del> | <del>+0.30</del> | <del>+0.4¢</del> |
|------------------------------|--------------------------|---------------|-----------------|-------------------|------------------|-----------------|------------------|--------------------------|-------------------------|-------------------------|
| <del>0.0</del>               |                          |               |                 |                   |                  | <del>100%</del> |                  |                          |                         |                         |
| <del>0.1</del>               |                          |               |                 | <del>6%</del>     | <del>24%</del>   | <del>40%</del>  | <del>24%</del>   | <del>6%</del>            |                         |                         |
| <del>0.15</del>              |                          | <del>3.</del> | <del>5%</del>   | <del>11%</del>    | <del>22%</del>   | <del>27%</del>  | <del>22%</del>   | <del>11%</del>           | <del>3.5%</del>         |                         |
| <del>0.2</del>               | <del>2.8%</del>          | <del>6.</del> | <del>7%</del>   | <del>12%</del>    | <del>18%</del>   | <del>20%</del>  | <del>18%</del>   | <del>12%</del>           | <del>6.7%</del>         | <del>2.8%</del>         |
| Bin                          | σagg=0 | 0.1    | 0.2      | 0.3        |                  |                 |                  |                          |                         |                         |
| µаgg -0.6ф |                          |               |                 | 1.9        |                  |                 |                  |                          |                         |                         |
| µаgg -0.5ф |                          |               | 0.9      | 3.4        |                  |                 |                  |                          |                         |                         |
| μ agg -0.4φ       |                          |               | 2.7      | 5.6        |                  |                 |                  |                          |                         |                         |
| μ agg -0.3φ       |                          |               | 6.5      | 8.3        |                  |                 |                  |                          |                         |                         |
| μagg -0.2φ |                          | 6      | 12       | 11.0       |                  |                 |                  |                          |                         |                         |
| μagg -0.1φ |                          | 24     | 18       | 13.0       |                  |                 |                  |                          |                         |                         |
| µagg       | 100               | 40     | 20       | 13.7       |                  |                 |                  |                          |                         |                         |
| μ agg +0.1¢       | 2                        | 24            | 18       | 13.0       |                  |                 |                  |                          |                         |                         |
| μagg +0.2  |                          | 6      | 12       | 11.0       |                  |                 |                  |                          |                         |                         |
| μ agg +0.3¢       | 2                        |               | 6.5      | 8.3        |                  |                 |                  |                          |                         |                         |
| μagg +0.4  | 2                        |               | 2.7      | 5.6        |                  |                 |                  |                          |                         |                         |
| μ agg +0.5¢       | 2                        |               | 0.9      | 3.4        |                  |                 |                  |                          |                         |                         |
| μ agg +0.6¢       |                          |               |                 | 1.9        |                  |                 |                  |                          |                         |                         |

|                  | Field Code Changed |   |
|------------------|--------------------|---|
|                  | Formatted          |   |
| $\ $             | Formatted          |   |
|                  | Formatted          |   |
| [[]]]            | Formatted          |   |
|                  | Formatted          |   |
| $\parallel \mid$ | Formatted          |   |
|                  | Formatted          |   |
|                  | Formatted          |   |
| 1                | Formatted          |   |
| $\parallel$      | Formatted          |   |
|                  | Formatted          |   |
|                  | Formatted          |   |
| Γ                | Formatted          |   |
| _                | Formatted          |   |
|                  | Formatted          |   |
|                  | Formatted          | ) |
| $\overline{)}$   | Formatted          |   |
|                  | Formatted          |   |
| $\sum$           | Formatted          |   |
| //               | Formatted          |   |
| 1                | Formatted          |   |
| 11,              | Formatted          |   |
| 1/               | Formatted          | ) |
| $\left  \right $ | Formatted          |   |
| /                | Formatted          |   |
|                  | Formatted          |   |
|                  |                    |   |

Table 5: Atmospheric temperature profiles during the eruptions at Mount St. Helens, Crater Peak (Spurr), Ruapehu, and Redoubt volcanoes. Profile for Mount St. Helens is for 18 May

890 1980, 1800 UTC, interpolated to the location of Ritzville, Washington (47.12°N, 118.38°W).

For Crater Peak (Spurr) the profile is for 17 September 1992, 1200 UTC, interpolated to the

892 location of Palmer, Alaska (61.6°N, 149.11°W). For Ruapehu the temperature profile is for

893 17 June 1996, 0000 UTC, interpolated to the location of Ruapehu. For Redoubt the sounding

was for 23 March 2009, 1200 UTC, at 62°N, 153°W. All soundings were taken from using
 RE1 reanalysis data at <a href="http://ready.arl.noaa.gov/READYamet.php">http://ready.arl.noaa.gov/READYamet.php</a>. 
[revised manuscript text omitted]